# Lmo2 expression defines tumor cell identity during T-cell leukemogenesis

Idoia García-Ramírez[1,2,†], Sanil Bhatia[3,†], Guillermo Rodríguez-Hernández[1,2], Inés González-Herrero[1,2], Carolin Walter[4], Sara González de Tena-Dávila[1,2], Salma Parvin[5,6], Oskar Haas[7], Wilhelm Woessmann[8], Martin Stanulla[9], Martin Schrappe[10], Martin Dugas[4], Yasodha Natkunam[11], Alberto Orfao[2,12], Verónica Domínguez[13], Belén Pintado[13], Oscar Blanco[2,14], Diego Alonso-López[15], Javier De Las Rivas[2,16], Alberto Martín-Lorenzo[1,2] ID, Rafael Jiménez[2,17], Francisco Javier García Criado[2,18], María Begoña García Cenador[2,18], Izidore S Lossos[5,6], Carolina Vicente-Dueñas[2,*,‡] ID, Arndt Borkhardt[3,**,‡] ID, Julia Hauer[3,***,‡] ID & Isidro Sánchez-García[1,2,****,‡] ID

## Abstract

The impact of LMO2 expression on cell lineage decisions during T-cell leukemogenesis remains largely elusive. Using genetic lineage tracing, we have explored the potential of LMO2 in dictating a T-cell malignant phenotype. We first initiated LMO2 expression in hematopoietic stem/progenitor cells and maintained its expression in all hematopoietic cells. These mice develop exclusively aggressive human-like T-ALL. In order to uncover a potential exclusive reprogramming effect of LMO2 in murine hematopoietic stem/progenitor cells, we next showed that transient LMO2 expression is sufficient for oncogenic function and induction of T-ALL. The resulting T-ALLs lacked LMO2 and its target-gene expression, and histologically, transcriptionally, and genetically similar to human LMO2-driven T-ALL. We next found that during T-ALL development, secondary genomic alterations take place within the thymus. However, the permissiveness for development of T-ALL seems to be associated with wider windows of differentiation than previously appreciated. Restricted Cre-mediated activation of *Lmo2* at different stages of B-cell development induces systematically and unexpectedly T-ALL that closely resembled those of their natural counterparts. Together, these results provide a novel paradigm for the generation of tumor T cells through reprogramming *in vivo* and could be relevant to improve the response of T-ALL to current therapies.

**Keywords** cancer initiation; epigenetic priming; mouse models; oncogenes; stem cells

**Subject Categories** Cancer; Development & Differentiation; Immunology

**The EMBO Journal (2018) 37: e98783**

1 Experimental Therapeutics and Translational Oncology Program, Instituto de Biología Molecular y Celular del Cáncer, CSIC-USAL, Salamanca, Spain
2 Institute of Biomedical Research of Salamanca (IBSAL), Salamanca, Spain
3 Department of Pediatric Oncology, Hematology and Clinical Immunology, Medical Faculty, Heinrich-Heine University Dusseldorf, Dusseldorf, Germany
4 Institute of Medical Informatics, University of Muenster, Muenster, Germany
5 Division of Hematology-Oncology, Department of Medicine, Sylvester Comprehensive Cancer Center, University of Miami, Miami, FL, USA
6 Department of Molecular and Cellular Pharmacology, Sylvester Comprehensive Cancer Center, University of Miami, Miami, FL, USA
7 Children's Cancer Research Institute, St Anna Children's Hospital, Vienna, Austria
8 Department of Pediatric Hematology and Oncology, Justus-Liebig-University Giessen, Giessen, Germany
9 Pediatric Hematology and Oncology, Hannover Medical School, Hannover, Germany
10 Department of Pediatrics, Christian-Albrechts-University of Kiel and University Medical Center Schleswig-Holstein, Kiel, Germany
11 Department of Pathology, Stanford University School of Medicine, Stanford, CA, USA
12 Servicio de Citometría y Departamento de Medicina, Universidad de Salamanca, Salamanca, Spain
13 Transgenesis Facility CNB-CBMSO, CSIC-UAM, Madrid, Spain
14 Departamento de Anatomía Patológica, Universidad de Salamanca, Salamanca, Spain
15 Bioinformatics Unit, Cancer Research Center (CSIC-USAL), Salamanca, Spain
16 Bioinformatics and Functional Genomics Research Group, Cancer Research Center (CSIC-USAL), Salamanca, Spain
17 Departamento de Fisiología y Farmacología, Edificio Departamental, Universidad de Salamanca, Salamanca, Spain
18 Departamento de Cirugía, Universidad de Salamanca, Salamanca, Spain
*Corresponding author. Tel: +34 923294813; E-mail: cvd@usal.es
**Corresponding author. Tel: +49 211 81 17680; E-mail: Arndt.Borkhardt@med.uni-duesseldorf.de
***Corresponding author. Tel: +49 211 81 17680; E-mail: Julia.Hauer@med.uni-duesseldorf.de
****Corresponding author. Tel: +34 923294813; E-mail: isg@usal.es
†These authors contributed equally to this work as first authors
‡These authors contributed equally to this work as senior authors

## Introduction

The identification of the cell-of-origin from which acute lymphoblastic leukemia (ALL) initially arises is of great importance, both for our understanding of the basic biology of tumors and for the translation of this knowledge to the prevention, treatment, and precise prognosis of ALL (Visvader, 2011). Traditionally, the identity of the cell-of-origin was extrapolated from the immunophenotypic characterization of a leukemic cell. However, several transcriptome studies have shown that the molecular characteristics of leukemic cells do not correspond, in many cases, to what they seem to be according to their immunophenotype (Lim *et al*, 2009; Gilbertson, 2011). For this reason, extrapolating the identity of the cancer cell-of-origin from the ALL phenotype, without appropriate functional lineage tracing, can lead to the wrong conclusions (Molyneux *et al*, 2010).

Lmo2 is one of the most frequent drivers of childhood T-ALL (Van Vlierberghe *et al*, 2006; Liu *et al*, 2017). LMO2 serves as a T-cell oncogene, recurrently translocated in T-ALL, and is implicated in leukemogenesis among X-linked severe combined immunodeficiency (SCID) patients, who received retroviral *IL2Rγc* gene therapy (Hacein-Bey-Abina *et al*, 2003, 2008; Pike-Overzet *et al*, 2007; Howe *et al*, 2008). Aberrant expression of *LMO2* in hematopoietic stem/progenitor cells (HSC/PC) or in immature T cells (present in the thymus) leads to thymocyte self-renewal, early lymphoid precursor's accumulation, and transformation to T-ALL (McCormack *et al*, 2010; Treanor *et al*, 2011; Cleveland *et al*, 2013; Chambers & Rabbitts, 2015). Moreover, *LMO2* was recently identified as one of the six transcription factors required for reprogramming committed murine blood cells into induced hematopoietic stem cells (Riddell *et al*, 2014). Notably, in addition to T-ALL, *LMO2* is expressed in hematologic cancer of the B-cell lineage including DLBCL (Natkunam *et al*, 2007; Cubedo *et al*, 2012) and BCP-ALL (de Boer *et al*, 2011; Malumbres *et al*, 2011; Deucher *et al*, 2015). Induction of pluripotency in blood cells and *LMO2* expression in B-cell malignancies suggest that *LMO2* might exert leukemogenic potential in specific hematopoietic cell lineages other than the T-cell lineage. Besides that, a significant proportion of human T-ALL displays rearrangements of immunoglobulin heavy-chain genes, which additionally supports this hypothesis (Mizutani *et al*, 1986; Szczepanski *et al*, 1999; Meleshko *et al*, 2005). However, despite frequent alterations of *Lmo2* in hematologic tumors, its impact on lineage organization during leukemogenesis and the importance of the cell-of-origin for heterogeneity and aggressiveness of Lmo2-driven tumors have remained unclear. By using *in vivo* genetic lineage tracing, we show that *Lmo2* expression in HSC/PC as well

as a precursor and mature B cells causes reprogramming and induction of T-ALL. Thereby the differentiation state of the tumor cell-of-origin influences the frequency and latency of T-ALL. These findings unveil a novel role of *Lmo2* expression and demonstrate that *Lmo2* promotes tumorigenesis in a manner contrasting that of other traditional oncogenes, which are persistently active in fully evolved tumor cells (Weinstein, 2002).

## Results

### Generation of a targeted mouse line conditionally expressing *Lmo2* in HSCs

Cell type-specific conditional activation of *Lmo2* is a powerful tool for investigating the cell-of-origin of T-ALL. To achieve this aim, the *Lmo2* cDNA was targeted to the ubiquitously expressed *Rosa26* locus (Mao *et al*, 1999) where the green fluorescent protein (eGFP) was linked to the mouse *Lmo2* cDNA via an internal ribosomal entry site (IRES). In the absence of Cre, neither *Lmo2* nor *eGFP* is expressed (Appendix Fig S1A and B).

Two sets of observations suggest a reprogramming effect of non-T-cell lineage cells by LMO2. First, *LMO2* expression due to retroviral insertion and transactivation in $CD34^+$ HSCs of X-SCID patients caused T-ALL but no other hematopoietic tumors (Hacein-Bey-Abina *et al*, 2008; Howe *et al*, 2008). And second, *Lmo2* expression in murine blood cells negatively regulated erythroid differentiation (Visvader, 2011) and gives rise to induced pluripotent stem (iPS) cells (Batta *et al*, 2014; Riddell *et al*, 2014). We thus aimed to model the capability of *Lmo2* to reprogram HSCs. Therefore, we initially crossed the *Rosa26-Lmo2* mice with a *Sca1-Cre* mouse strain (Mainardi *et al*, 2014), in order to initiate *Lmo2* expression in HSCs and maintain its expression in all hematopoietic cells (Appendix Fig S1C). Young *Rosa26-Lmo2 + Sca1-Cre* mice showed regular hematopoietic cell differentiation in the bone marrow, peripheral blood, spleen, and thymus (Appendix Figs S1C–E and S2A–D). *Rosa26-Lmo2 + Sca1-Cre* mice had a shorter lifespan than their *wild-type* (WT) littermates [Fig 1A; *P* < 0.0001; log-rank (Mantel–Cox) test] due to the development of T-ALL (96.7%; 30/31) that manifested as thymoma, splenomegaly, and disrupted thymic, liver, and splenic architectures (Fig 1B; Appendix Fig S3A and B). Fluorescent activating cell sorting (FACS) analysis of leukemic cells revealed an immature $CD8^+CD4^{+/-}$ cell surface phenotype (Fig 1C; Appendix Fig S3C) with Lmo2 expression in the tumor T cells (Fig 1D) and clonal immature T-cell receptor (TCR) rearrangement

**Figure 1. T-ALL development in *Rosa26-Lmo2 + Sca1-Cre* mice.**

A  Leukemia-specific survival of *Rosa26-Lmo2 + Sca1-Cre* mice (red line, *n* = 31), showing a significantly (log-rank ***P* < 0.0001) shortened lifespan compared to control littermate WT mice (black line, *n* = 20) as a result of T-ALL development.

B  An example of thymomas observed in the *Rosa26-Lmo2 + Sca1-Cre* mice studied. A thymus from a control littermate WT mouse is shown for reference. Hematoxylin and eosin staining showing infiltration of the thymus in *Rosa26-Lmo2 + Sca1-Cre* leukemic mice. Images are photographed at 400× magnification (scale bars: 200 μm).

C  *GFP* expression in the pre-leukemic and leukemic cells from *Rosa26-Lmo2 + Sca1-Cre* mice, respectively. A control littermate *WT* mouse is shown for reference.

D  Western blot analysis for Lmo2 and actin in T cells from the thymus of a *wild-type* mouse (1) and from the thymus of a *Rosa26-Lmo2 + Sca1-Cre* leukemic mouse (2). Tumoral cells of *Rosa26-Lmo2 + Sca1-Cre* T-ALL showed expression of the Lmo2 protein.

E  TCR clonality in *Rosa26-Lmo2 + Sca1-Cre* mice. PCR analysis of TCR gene rearrangements in infiltrated thymuses of diseased *Rosa26-Lmo2 + Sca1-Cre* leukemic mice. Sorted DP T cells from the thymus of healthy mice served as a control for polyclonal TCR rearrangements. Leukemic thymus shows an increased clonality within their TCR repertoire (indicated by the code number of each *Rosa26-Lmo2 + Sca1-Cre* mouse analyzed).

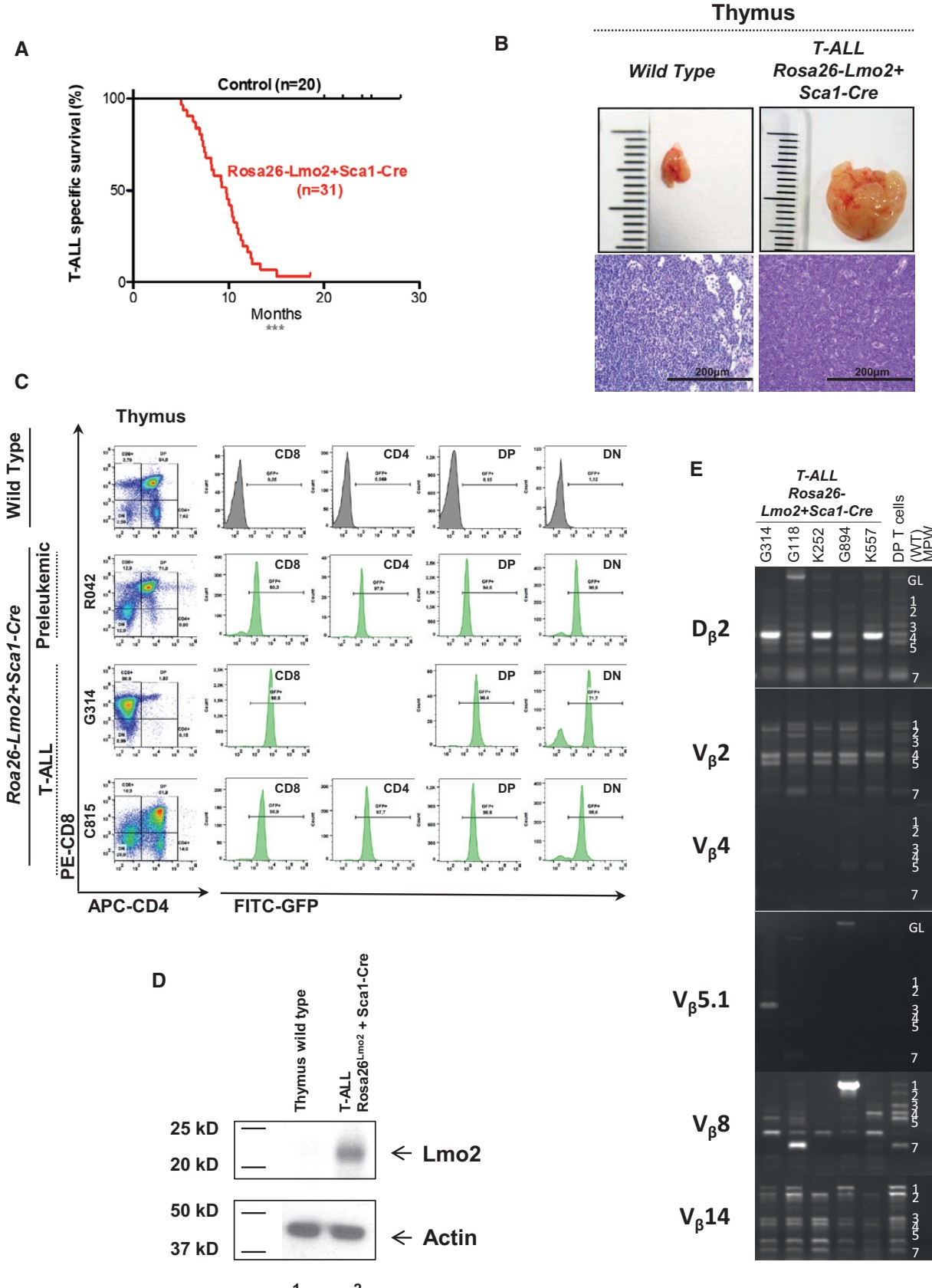

**Figure 1.**

(Fig 1E). We also performed whole-exome sequencing (WES) of these Lmo2$^+$ T-ALLs ($n = 9$; Table 1), which were derived from thymuses of diseased *Rosa26-Lmo2 + Sca1-Cre* mice. We detected 23 somatic mutations, including six mutations in genes recorded in the cancer gene list (Table 1; Table EV1). Briefly, we identified recurrent *Notch1* single-nucleotide variations (SNVs; 3/9) and *Notch1* indels (4/9), *Kras* SNVs (3/9), and *Nras* SNVs (1/9; Table 1). This model corroborated previous findings, especially the observation from the SCID-X1 gene therapy trial, where integration of γC vector occurred close or in the LMO2 locus and *Lmo2* expression was maintained throughout the progeny of the targeted cell (Hacein-Bey-Abina *et al*, 2003, 2008; Pike-Overzet *et al*, 2007; Howe *et al*, 2008). However, in our model *Lmo2* expression was maintained constitutively, not only in HSC/PC but also in precursor and mature

T cells (McCormack *et al*, 2010). Thus, a definite conclusion about an exclusive reprogramming effect of *Lmo2* in murine HSC/PC in contrast to its expression in T-cell precursors and mature T cells was limited.

### *Lmo2* functions as a "hit-and-run" oncogene in T-ALL development

We next addressed these limitations and modeled the scenario of HSC/PC restricted *Lmo2* expression *in vivo* in a mouse strain where *Lmo2* expression was initiated and maintained only in HSC/PC by placing *Lmo2–TdTomato* cDNA (Shaner *et al*, 2004) under the control of the stem-cell-specific *Sca1* promoter (*Sca1-Lmo2*; Appendix Fig S4A). All T-cell subsets in the thymus contained a

**Table 1. Recurrent mutations in mouse models and human Lmo2$^+$ T-ALL.**

The 28 frequently mutated targets either in different mouse models or in our human T-ALL cohort with their overlap to the most (55) common targets mutated in T-ALL, described by Liu *et al* (2017). The left panel shows genes with mutations, whereas the numbers at the top are depicting the different mouse or human samples. Every box is specifying the numbers of mouse or human samples sequenced. All the mutations displayed are confirmed by Sanger sequencing.

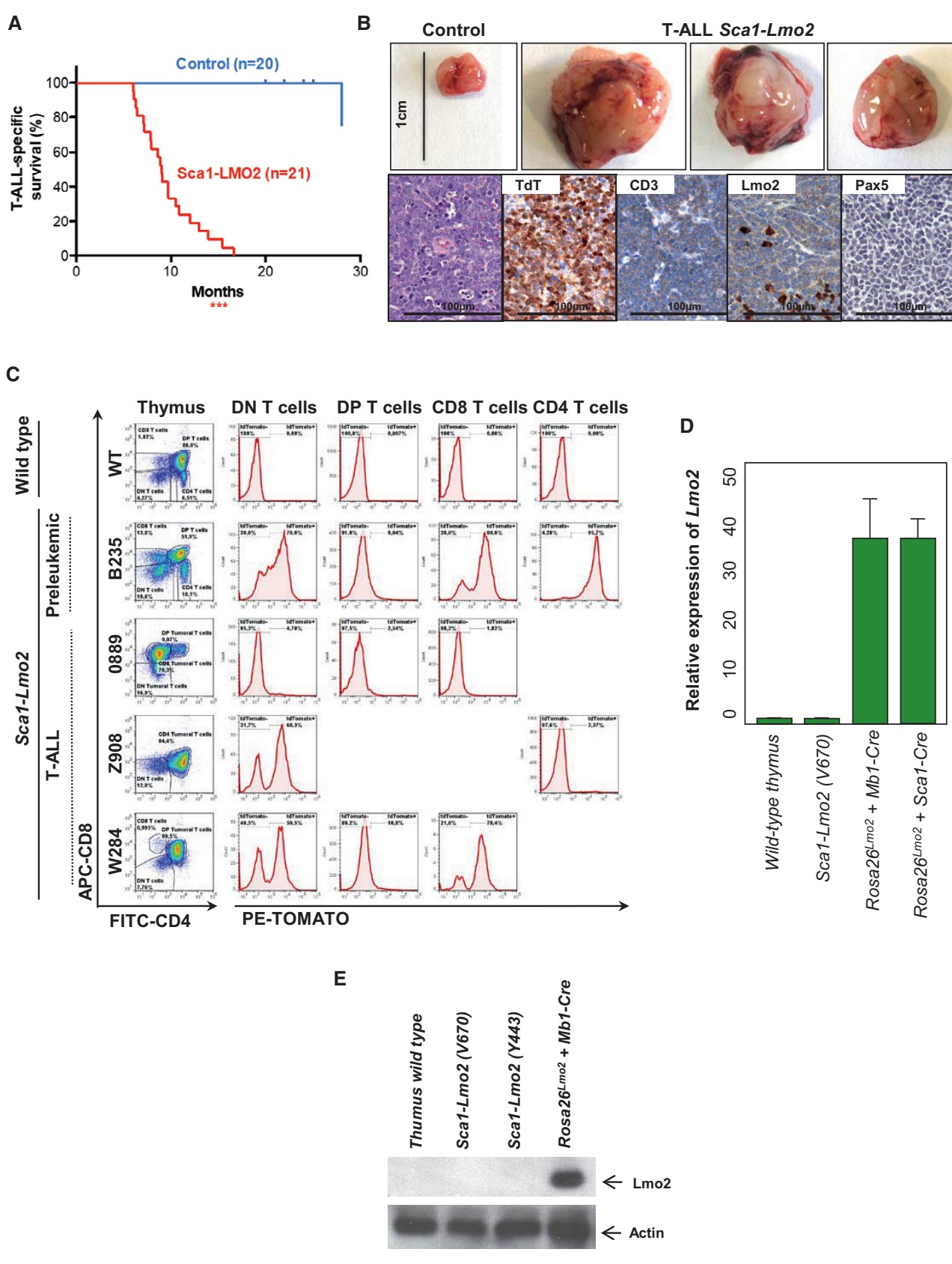

**Figure 2.**

◀

**Figure 2.  Reprogramming of HS/PCs cells to aggressive malignant mature T cells.**

C   Flow cytometric analysis of T-cell subsets in the thymuses of diseased *Sca1-Lmo2* mice. Representative plots of cell subsets from the thymuses are shown. These exhibited the accumulation of DP, CD8, or CD4 single-positive tumoral T cells. Thymuses from a control littermate WT mouse and a pre-leukemic *Sca1-Lmo2* mouse are shown for reference. Flow cytometric images are representative of 17 mice analyzed. Tracking of the TdTomato marker for *Lmo2* transgene expression in the thymomas of *Sca1-Lmo2* mice shows that tumor T cells are TdTomato-negative in 100% (17/17) of the *Sca1-Lmo2* mice studied. However, a mosaic of *Lmo2* expression remains present within non-tumoral T-cell populations (not denoted as tumoral populations). Three plots of cell subsets from the thymuses of diseased *Sca1-Lmo2* mice are shown and are representative of the analysis of 17 diseased *Sca1-Lmo2* mice. *TdTomato* expression in the thymuses of a control littermate WT mouse and a pre-leukemic *Sca1-Lmo2* mouse is shown for reference.

D   Relative expression of *Lmo2* in sorted-purified leukemic *Sca1-Lmo2* cells compared to control thymus *wild-type* cells. Leukemic cells of both *Rosa26-Lmo2 + Mb1-Cre* and *Rosa26-Lmo2 + Sca1-Cre* mice were used as a positive control. The fold change in each group, calculated as $2^{-\Delta\Delta C_t}$ sample, was compared. Bars represent the mean and the standard deviation of three replicates.

E   Western blot analysis for Lmo2 and actin in T cells from a *wild-type* thymus (1), in sorted-purified leukemic *Sca1-Lmo2* cells (2, 3) and from the thymus of a *Rosa26-Lmo2 + Mb1-Cre* leukemic mouse (4).

mosaic of *Lmo2* expression (Appendix Fig S4B). *Sca1-Lmo2* mice showed the regular distribution of hematopoietic populations in early post-gestational development, with *TdTomato* expression in all hematopoietic cell lineages (Appendix Fig S4C–G). By 3 months, a decrease in the double-positive (DP) T-cell population was accompanied by an increase in pre-leukemic double-negative (DN) T cells and CD8 T cells (Appendix Fig S4H). *Lmo2* expression was detected by quantitative polymerase chain reaction (qPCR); enhanced expression of *Cdkn2a* was observed in the thymus in transgenic mice, consistent with the induction of Lmo2-dependent oncogenic stress (Appendix Fig S4I). The Sca1 promoter is active in subsets of T-cell precursors, and thus, both *Lmo2*-expressing and *Lmo2*-non-expressing precursor T cells coexisted in the thymus (Appendix Fig S4J). Studying whether the T-ALL cases are Lmo2-Tomato-positive or Lmo2-Tomato-negative has allowed identifying whether the *Lmo2* expression is needed for the survival of T-ALL cells (Tomato$^+$) or it serves as an earlier reprogramming event in leukemogenesis (Tomato$^-$).

Sca1-Lmo2 mice had a shorter lifespan than their wild-type (WT) littermates due to a highly disseminated form of T-ALL, consisting of a clonally immature CD8 or CD4 single-positive/DP-like population (Fig 2A–C; Appendix Fig S5A–E), as reported for human T-ALL (Van Vlierberghe *et al*, 2006) and *Rosa26-Lmo2 + Sca1-Cre* mice (Fig 1). Histological thymus sections were characterized by infiltrates of highly proliferative tumors and CD3 and TdT positivity (Fig 2B). Surprisingly, all *Sca1-Lmo2* T-ALL cases studied (18 out of 21) were *TdTomato*$^-$. Because there is evidence to suggest that the immunogenicity and cytotoxicity of the fluorescent marker potentially may confound the interpretation of *in vivo* experimental data (Ansari *et al*, 2016), we next formally excluded the possibility that the cells that were originally marked with the fluorescent marker cannot be accurately traced over time by showing that tumors had lost their *Lmo2* expression by three different complementary approaches: immunohistochemistry (Fig 2C) and both real-time PCR and Western blot in sorted-purified leukemic *Sca1-Lmo2* cells (Fig 2D and E). This observation indicates that an early expression of the *Lmo2* oncogene in HSC/PC has the potential to induce aggressive T-ALL without any need for its perpetual expression to develop T-ALL.

## Tumor T cells in *Sca1-Lmo2* mice display genetic signatures analogous to human malignant T cells

In human and mouse T-ALL, genomic gains and losses reflect genomic instability (Maser *et al*, 2007; Hacein-Bey-Abina *et al*, 2008; Howe *et al*, 2008; De Keersmaecker *et al*, 2010). We analyzed DNA from leukemic cells using array-comparative genomic hybridization (aCGH). Twelve *Sca1-Lmo2* leukemias were analyzed and revealed copy number loss of *Cdkn2a/b* (2/12) and *Bcl11b* (4/12), similar to human T-ALL (Diccianni *et al*, 1997; Gutierrez *et al*, 2011), as well as c-*Myc* amplification (8/12; Fig 3A). Hence, T-cell progenitors lacking in *Lmo2* expression were genomically unstable and were clonally selected; moreover, they have acquired additional aberrations. To explore the relevance of our findings for human T-ALL, we analyzed molecular expression signatures in thymic Lmo2-negative T-ALL cells (*Sca1-Lmo2*; Fig 3B). We observed upregulation of the Notch1 pathway (Weng *et al*, 2004) and c-Myc transcriptional targets (Weng *et al*, 2006), as well as downregulation of *Fbxw7*, *Pten* (O'Neil *et al*, 2007; Palomero *et al*, 2007; Thompson *et al*, 2007; Van Vlierberghe & Ferrando, 2012), *Cyld*, and *Cdkn1b* (Komuro *et al*, 1999; Dohda *et al*, 2007; Espinosa *et al*, 2010; D'Altri *et al*, 2011; Fig 3B and C). Hence, similar oncogenic pathways were deregulated in murine T-ALL arising from *Lmo2*-negative T cells, which is consistent with human T-ALL but not with B-ALL. Thus, we can exclude that the correlation between murine and human T-ALL reflects a transformed state.

Gene sets pertaining to stem cell identity were highly enriched in Lmo2-negative T-ALL (Fig 3C), suggesting that the stem-cell-specific transcriptional program remains activated in the absence of *Lmo2* expression. However, gene expression analysis identified upregulation of the Notch1 pathway in thymic pre-leukemic tomato$^-$ versus tomato$^+$ cells, comparable to the leukemic T cells (Fig 3D and E; Appendix Fig S6A–E; Table EV2). Thus, these data suggest that *Lmo2* initiates a reprogramming-like mechanism in HSC/PC, while the T-ALL is maintained independently of Lmo2 expression.

To further explore the relevance of our findings to human T-ALL, we performed WES of 10 Lmo2-negative tumors from the thymuses of diseased *Sca1-Lmo2* mice (Table 1). We detected 40 somatic mutations, including 10 mutations in genes recorded in the cancer

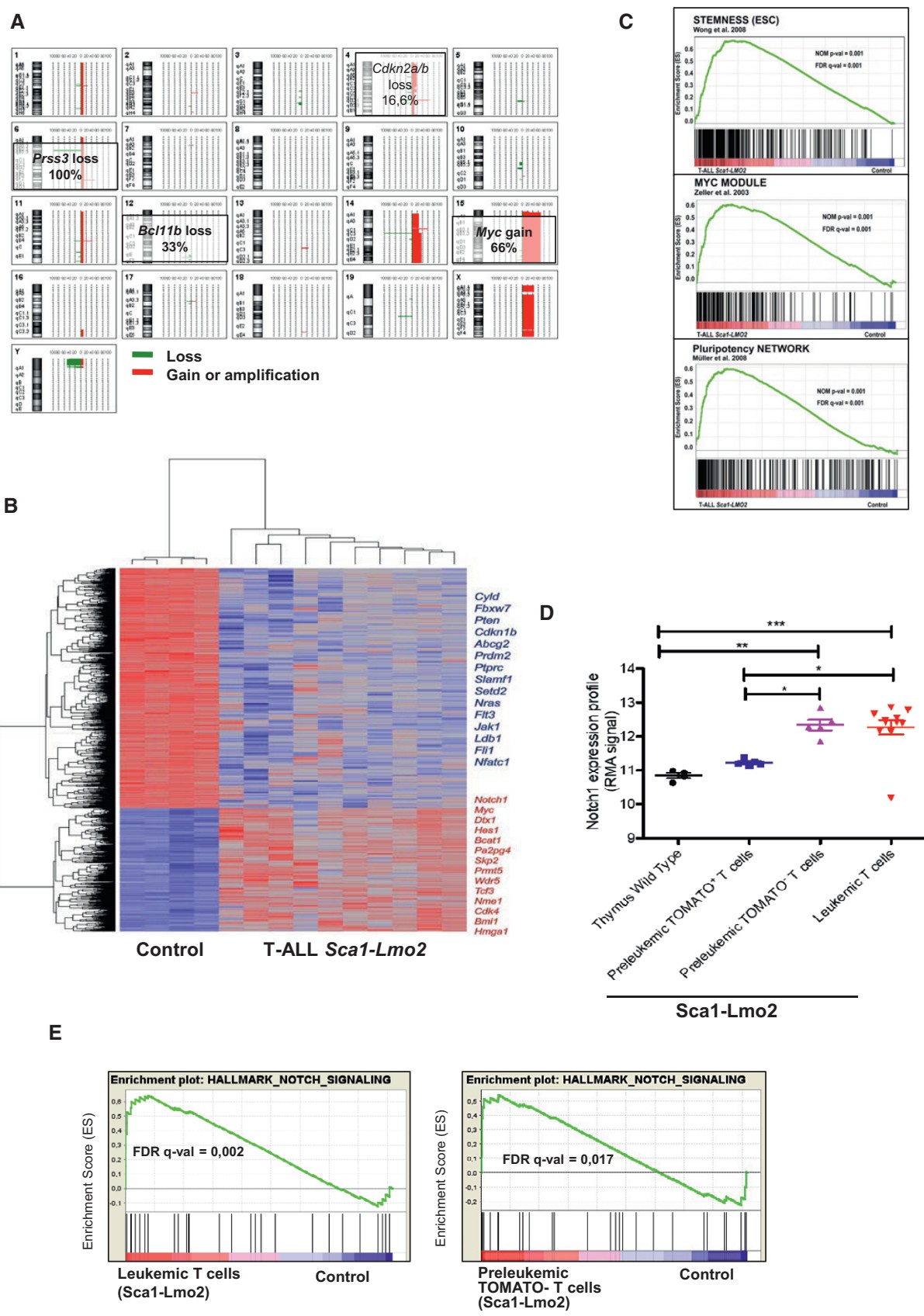

Figure 3.

◄

**Figure 3.  Molecular identity of tumor cells in *Sca1-Lmo2* T-ALL.**

A   Overview of chromosomal imbalances mapped by 4x180k oligonucleotide aCGH in 12 T-ALL cases in *Sca1-Lmo2* mice. The 20 chromosome ideograms of T-ALL *Sca1-Lmo2* mice are shown with DNA deletions drawn as green lines and amplifications or gains as red lines. Selected chromosomal alterations are highlighted.

B   Genes significantly induced or repressed within tumor T cells of *Sca1-Lmo2* mice in comparison with WT littermates, as determined by significance analysis of microarrays using FDR 1%. Each row represents a separate gene, and each column denotes a separate mRNA sample. The level of expression of each gene in each sample is represented using a red–blue color scale (upregulated genes are displayed in red and downregulated genes in blue). Selected genes are highlighted.

C   GSEA of the transcriptional signatures within tumor T cells compared with control WT littermates. Gene expression data from *Sca1-Lmo2* tumor T cells showed significant enrichment in embryonic stem cell genes (Wong *et al*, 2008) (GSEA FDR *q*-value = 0.001), Myc target genes (Zeller *et al*, 2003) (GSEA FDR *q*-value = 0.001), and pluripotency genes (Muller *et al*, 2008) (GSEA FDR *q*-value=0.001).

D   Notch1 expression profile in control WT T cells, pre-leukemic tomato⁺ T cells, pre-leukemic tomato⁻ T cells, and leukemic T cells. The statistical test used was Mann–Whitney *U*-test: wild-type thymus versus leukemic T cells (\*\*\**P* < 0.001), wild-type thymus versus pre-leukemic tomato⁻ T cells (\*\**P* = 0.0031), pre-leukemic tomato⁺ T cells versus leukemic T cells (\**P* = 0.0127), and pre-leukemic tomato⁺ T cells versus pre-leukemic tomato⁻ T cells (\**P* = 0.0159). Error bars represent the mean ± SEM.

E   GSEA of the Notch signaling in leukemic and tomato⁻ cells (GSEA FDR *q*-value = 0.002, 0.017, respectively).

gene list (Table EV1); primarily, we observed recurrent *Notch1* (SNVs 5/10, indels 5/10) and *KRas* (SNVs 1/10) mutations (Table 1), consistent with *Rosa26-Lmo2 + Sca1-Cre* and human *LMO2*⁺ T-ALL pathogenesis (Table 1). However, we did not identify *Lmo2* target genes and/or pathways that could replace *Lmo2* function in Lmo2-negative T-ALL. WES of a corresponding human LMO2⁺ T-ALL cohort (*n* = 9) in which translocation t(11;14)(p13; q11) was confirmed by fluorescent *in situ* hybridization (FISH) analysis (Table EV3; Appendix Fig S7) corroborated the relevance of these mutations. Briefly, we found 34 somatic alterations, including eight genes recorded in the cancer gene list, and confirmed recurrent *NOTCH1* (SNVs 6/9, indels 1/9), *KRAS* (SNVs 1/9), *NRAS* (SNVs 2/9), *FBXW7* (SNVs 2/9), and *MTOR* (SNVs 2/9) mutations (Table 1; Table EV1). Human LMO2⁺ and *Sca1-Lmo2* T-ALL showed highly recurrent SNVs and indels in *NOTCH1* (p.L1585P) and *KRAS* (p.G12D/V; Table 1), targeting the same amino acid. Thus, our data suggest that transient *Lmo2* expression in murine HSC/PCs is sufficient for induction of human-like T-ALL without the need for sustained *Lmo2* expression in the T-ALL bulk.

**Secondary genomic alterations take place within the thymus during T-ALL development**

In an attempt to exclude the impact of *Lmo2* expression in thymic precursor cells and to analyze specifically the reprogramming potential of *Lmo2* expression in HSC/PC, we crossed *Sca1-Lmo2* mice with thymus-deficient *nu/nu* mice. *Sca1-Lmo2 + nu/nu* mice had a similar lifespan compared to *Sca1-Lmo2* (Fig 4A). Leukemic *Sca1-Lmo2 + nu/nu* mice (*n* = 8/10) had enlarged spleens and succumbed to a highly disseminated form of leukemia that infiltrated both hematopoietic and non-hematopoietic tissues (Fig 4B and C; Appendix Fig S8A). In contrast to *Sca1-Lmo2* T-ALL, *Lmo2* was expressed in *Sca1-Lmo2 + nu/nu* leukemia (Fig 4C) and expression array data showed enrichment in human early T-cell precursor (ETP) ALL genes (Fig 4D), in agreement with human ETP ALL cases that commonly showed *LMO2/LYL1* deregulation (Liu *et al*, 2017). In addition, these leukemias showed enrichment in pluripotency, stemness (Fig 4D; Appendix Fig S8B and C), underscoring a reprogramming effect of *Lmo2* in HSC/PC. These results suggest that *Lmo2* is able to reprogram HSC/PC before entering the thymus. Next, we performed WES of five *Sca1-Lmo2 + nu/nu* mice with leukemia and observed 14 somatic alterations (SNVs) in *Cdh11* (1/5), *Cd1d1* (2/5), *Sept6* (2/5), and *Hspa1l* (1/5; Table 1; Table EV1). We did not observe *Notch1* or *Ras* indels/SNVs, in

contrast to T-ALL from *Rosa26-Lmo2 + Sca1-Cre* and *Sca1-tomato-IRES-Lmo2* mice and human LMO2⁺ T-ALL (Table 1). Hence, *Lmo2* is able to reprogram the cellular identity of HSC/PC into a tumorigenic one, but the thymus is indispensable to retain the T-ALL phenotype.

**B-cell-restricted *Lmo2* expression reprograms B cells into T-ALL**

We next asked whether the reprogramming potential of *LMO2* is restricted to the HSC/PC compartment or this ability applies to precursor and mature non-T-cell lineage cells. *Lmo2* is expressed in other types of hematologic cancer including diffuse large B-cell lymphoma (DLBCL; Natkunam *et al*, 2007; Cubedo *et al*, 2012) and B-cell precursor acute lymphoblastic leukemia (BCP-ALL; de Boer *et al*, 2011; Malumbres *et al*, 2011; Deucher *et al*, 2015), and a significant proportion of human T-ALL exhibits rearrangement of immunoglobulin heavy-chain genes (Mizutani *et al*, 1986; Szczepanski *et al*, 1999; Meleshko *et al*, 2005). Thus, we next address the effects of *Lmo2* expression in B cells. We initially used pro-B cells as targets for reprogramming because they carry genomic rearrangements of genes encoding VDJ regions of immunoglobulin heavy-chain locus that serve as natural genetic barcodes and they have weak barriers for reprogramming (Riddell *et al*, 2014). To this aim, we crossed the *Rosa26-Lmo2* mice with an *Mb1-Cre* mouse strain (Hobeika *et al*, 2006). The resulting strain deletes the stop cassette upon B-lineage commitment at the pro-B-cell level via the Cre recombinase, driven by the promoter from *Mb1* locus encoding the immunoglobulin-associated alpha chain Cd79a. FACS analysis confirmed uniform and efficient GFP expression at the pro-B stage, and therefore all subsequent stages of B-cell differentiation (Appendix Fig S9A). B cells from *Rosa26-Lmo2 + Mb1-Cre* mice showed a developmental pattern comparable to that of B cells from their control littermates (Appendix Fig S9B), which indicated that induction of Lmo2 at the pro-B-cell stage has a minimal effect on B-cell development. GFP expression was not detected outside the B-cell lineage in *Rosa26-Lmo2 + Mb1-Cre* mice as the frequency of GFP⁺ cells within both the BM myeloid progenitors and thymus T cells was undetectable (Appendix Fig S9A). These results also indicated that forced expression of Lmo2 was not able to reprogram committed progenitors of B cells into normal T lymphocytes. Importantly, *Rosa26-Lmo2 + Mb1-Cre* mice do not develop B-cell malignancies. However, *Rosa26-Lmo2 + Mb1-Cre* mice showed a shorter lifespan than their wild-type (WT) littermates (Fig 5A) due to the development of aggressive T-cell malignancies (5/27; 18.5%).

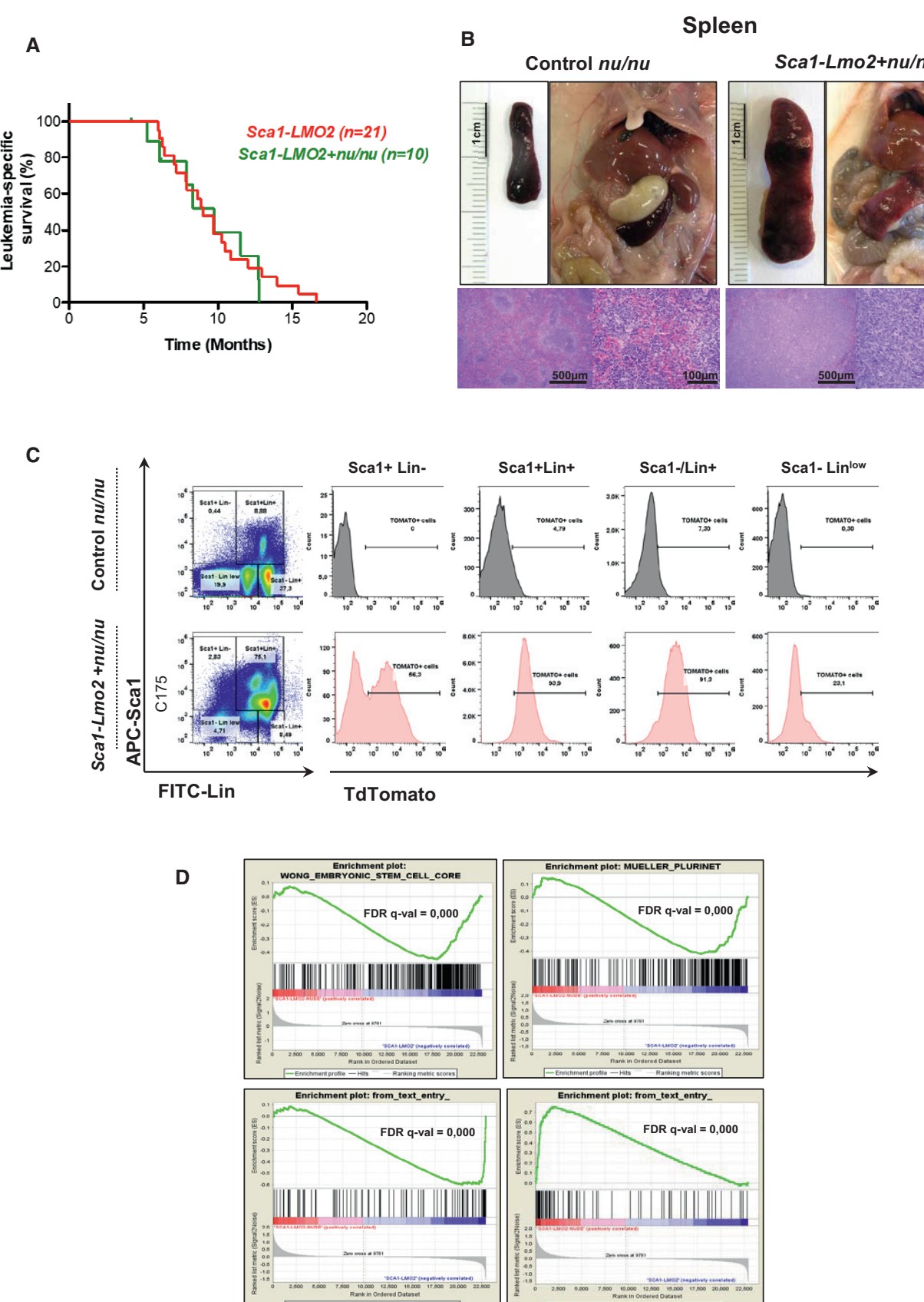

**Figure 4.**

◄

**Figure 4.  Leukemia development in *Sca1-Lmo2* + *nu/nu* mice.**

A  Leukemia-specific survival of *Sca1-Lmo2* + *nu/nu* mice (green line, *n* = 10), showing a similar shortened lifespan compared to *Sca1-Lmo2* mice (red line, *n* = 21) as a result of leukemia development.
B  An example of splenomegaly observed in *Sca1-Lmo2* + *nu/nu* mice studied pointing out by an arrowhead. Hematoxylin and eosin staining showing infiltration of spleen from *Sca1-Lmo2* + *nu/nu* leukemic mice. A spleen from a control littermate nu/nu mouse is shown for reference.
C  *TdTomato* expression in the leukemic cells from *Sca1-Lmo2* + *nu/nu* mouse. A control littermate nu/nu mouse is shown for reference.
D  GSEA of the transcriptional signatures within tumor cells of *Sca1-Lmo2* + *nu/nu* mice compared to tumor T cells of *Sca1-Lmo2* mice. Gene expression data from *Sca1-Lmo2* + *nu/nu* tumor cells showed significant enrichment of embryonic stem cell genes (GSEA FDR *q*-value = 0.000), pluripotency genes (GSEA FDR *q*-value = 0.000), genes upregulated in human ETP T-ALL (GSEA FDR *q*-value = 0.000), and genes downregulated in human ETP T-ALL (GSEA FDR *q*-value = 0.000).

Malignant T cells were primarily either double-positive for CD4/CD8 or single-positive for CD8 or single-positive for CD4 (Fig 5B), with Lmo2 expression in the tumor T cells (Fig 2D and E). These mice also showed infiltration of malignant cells into the spleen, liver, and thymus, resulting in disruption of normal architecture (Appendix Fig S9C). The latency of these *Rosa26-Lmo2* + *Mb1-Cre* T-ALL was higher than the latency of *Rosa26-Lmo2* + *Sca1-Cre* T-ALLs (Fig 5A), suggesting that the cell-of-origin impacts the disease malignancy.

Due to the increased expression of *Lmo2* in DLBCL (Natkunam *et al*, 2007; Cubedo *et al*, 2012), we therefore next crossed conditional *Rosa26-Lmo2* mice with the *Aid-Cre* strain, which expresses Cre recombinase in germinal center (GC) B cells (Crouch *et al*, 2007). Upon reaching immunological maturity, *Rosa26-Lmo2* + *Aid-Cre* mice were injected with T-cell-dependent antigen sheep red blood cells (SRBC) to induce *Lmo2* expression in GC cells. FACS analysis confirmed uniform and efficient GFP expression at GC stage (Appendix Fig S9D) and therefore within the subsequent stages of B-cell differentiation (Appendix Fig S9E). GFP expression was not detected in bone marrow progenitor B cells, bone marrow myeloid cells, and thymus T cells from pre-leukemic *Rosa26-Lmo2* + *Aid-Cre* mice (Appendix Fig S9E). These results also indicated that forced expression of Lmo2 within GC B cells was not able to contribute to normal T-cell development. However, *Lmo2* expression in GC cells did not result in DLBCL or other types of B-cell malignancies. However, 5% (1/19) of *Rosa26-Lmo2* + *Aid-Cre* mice developed aggressive T-cell malignancy (Fig 5C). Malignant T cells were primarily CD8$^+$CD4$^{+/-}$ (Fig 5D). These mice showed infiltration of malignant cells into the spleen, liver, kidney, and lung, resulting in disruption of normal architecture (Appendix Fig S9F). Similarly, the latency of the *Rosa26-Lmo2* + *Aid-Cre* T-ALL was even higher than the latency of *Rosa26-Lmo2* + *Sca1-Cre* and *Rosa26-Lmo2* + *Mb1-Cre* T-ALLs (Fig 5C), reinforcing the evidence that the cell-of-origin influences the disease malignancy.

T-ALL leukemia, which originated from either pro-B or GC cells, showed clonal TCR rearrangements (Fig 5E) and a significant similarity to *Rosa26-Lmo2* + *Sca1-Cre* tumors at the genomic level due to the presence of recurrent *Notch1* (SNVs (3/4), indels (1/4) in *Rosa26-Lmo2* + *Mb1-Cre*; SNVs (1/1), indels (1/1) in *Rosa-Lmo2* + *Aid-Cre*) mutations (Table 1; Table EV1). In line with a B-cell origin, *Rosa26-Lmo2* + *Mb1-Cre* and *Rosa26-Lmo2* + *Aid-Cre* T-ALL also showed clonal genomic rearrangements of genes encoding VDJ regions of immunoglobulin heavy-chain locus (Fig 5F). These results show that B-cell-restricted *Lmo2* expression can induce T-ALL in mice, a disease that never appears in control WT littermates. Furthermore, we show that the differentiation state of the cell-of-origin influences the frequency of T-ALL. Together, these data support a novel function of Lmo2 in mice, where the cell-of-origin

differentiation state does not dictate the Lmo2 tumor cell identity. Lmo2 expression in non-T-cell lineage cells including HSC/PC and B cells causes reprogramming with a common final path to T-ALL development.

## Transcriptomics landscape of Lmo2-driven T-ALL

To elucidate the differential transcriptomics landscape among different mouse models employed in this study, we next performed paired-end RNA-seq on *Rosa26-Lmo2* + *Sca1-Cre* (*n* = 3), *Sca1-Lmo2* (*n* = 4), *Rosa26-Lmo2* + *Mb1-Cre* (*n* = 2), *Rosa26-Lmo2* + *Aid-Cre* (*n* = 1), and WT-thymus (*n* = 4) mice. The 500 genes with the highest variance among the difference murine models were depicted (Fig 6A) with their corresponding FPKM values (Table EV5). Next, gene set enrichment analysis (GSEA) of *Rosa26-Lmo2* + *Sca1-Cre* and *Sca1-Lmo2* mouse-based gene signatures, against a human T-ALL childhood expression set with healthy controls (Mootha *et al*, 2003; Subramanian *et al*, 2005), was performed (Fig 6B). The upregulated *Rosa26-Lmo2* + *Sca1-Cre* signature shows a significant enrichment in the human T-ALL group which is in accordance with the human T-ALL situation wherein the expression of LMO2 is present throughout in tumor cells.

# Discussion

Understanding the stepwise events taking place during tumor cell evolution is difficult, because of many genetic alterations that become clonally selected by the time of clinically manifested T-ALL (Nowell, 1976). In principle, leukemogenesis is a process whereby a normal cell acquires novel but aberrant (malignant) identity in order to propagate a clonal population. This is only possible if the oncogenic event(s) have an inherent reprogramming capacity and the leukemia-initiating cell has the necessary plasticity (Sanchez-Garcia, 2015). Several prior studies have been involved in studying aberrations in HSC/PC as an important driver for myeloid and B-cell hematopoietic neoplasms through a reprogramming mechanism (Perez-Caro *et al*, 2009; Vicente-Duenas *et al*, 2012a,c; Green *et al*, 2014; Rodriguez-Hernandez *et al*, 2017), but to our knowledge, there is no evidence that a similar mechanism may be relevant for T-ALL. Two alternative explanations can be contemplated to interpret the close association existing between the *LMO2* oncogene and human T-ALL development: on the one side, the classical interpretation that considers that the role of *LMO2* as the T-ALL-initiating genetic alteration takes place in a committed/differentiated target T cell. Under this hypothesis, LMO2 is required for the immortalization of this committed target T cell that will later suffer additional genetic alterations with time which will further deregulate its

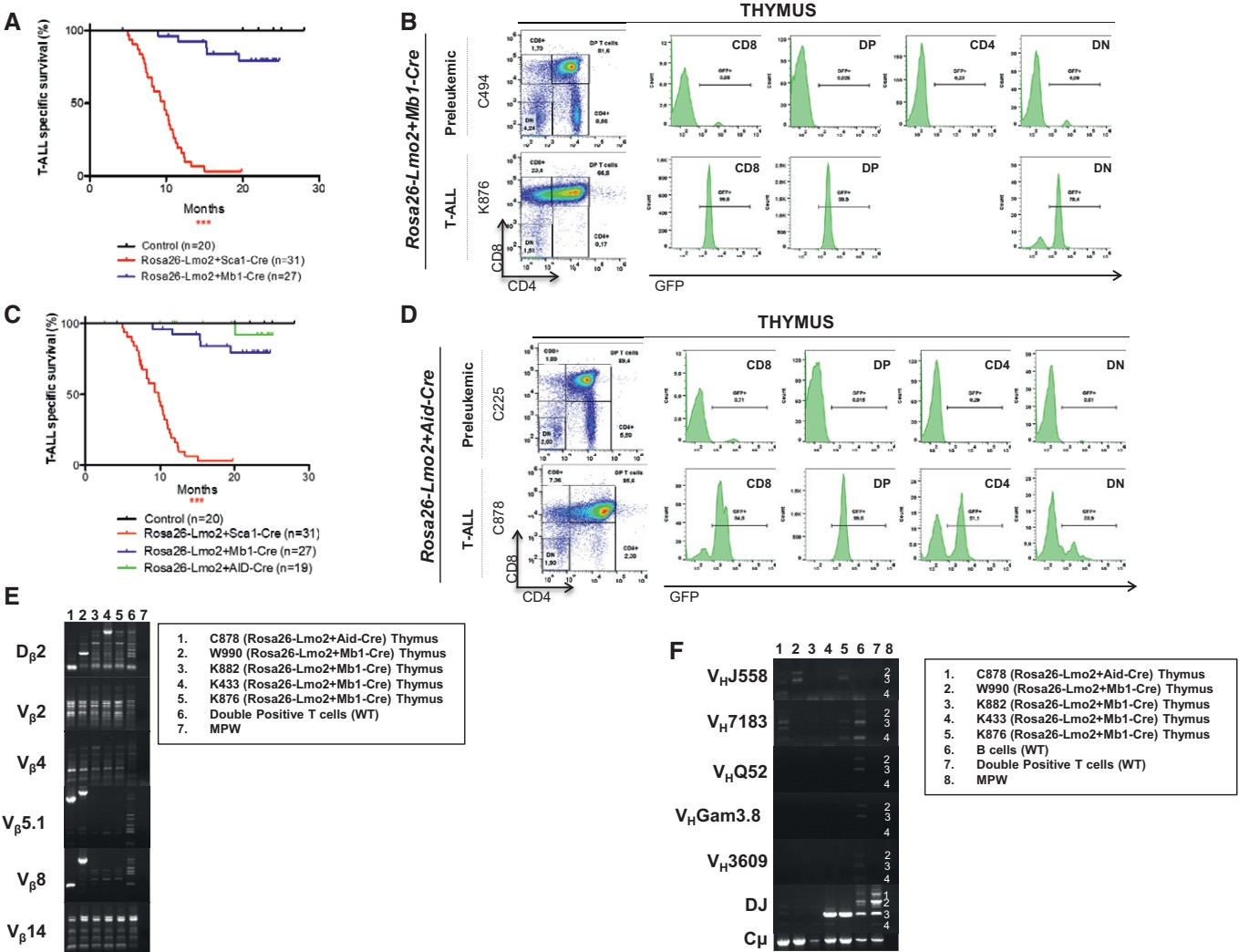

**Figure 5.  T-ALL development through *Lmo2* expression in B cells.**

A   Leukemia-specific survival of *Rosa26-Lmo2 + Mb1-Cre* mice (blue line, *n* = 27), showing a significantly (log-rank ***P < 0.0328) shortened lifespan compared to control littermate WT mice (black line, *n* = 20) as a result of T-ALL development. The latency of *Rosa26-Lmo2 + Mb1-Cre* T-ALL is higher than that of *Rosa26-Lmo2 + Sca1-Cre* T-ALL.

B   *GFP* expression in the pre-leukemic and leukemic cells from *Rosa26-Lmo2 + Mb1-Cre* mice, respectively.

C   Leukemia-specific survival of *Rosa26-Lmo2 + Aid-Cre* mice (green line, *n* = 19), not showing a significantly (log-rank ***P < 0.3173) shortened lifespan compared to control littermate WT mice (black line, *n* = 20). The latency of *Rosa26-Lmo2 + Aid-Cre* T-ALL is higher than that of *Rosa26-Lmo2 + Sca1-Cre* and *Rosa26-Lmo2 + Mb1-Cre* T-ALLs.

D   *GFP* expression in the pre-leukemic and leukemic cells from *Rosa26-Lmo2 + Aid-Cre* mice, respectively.

E   TCR clonality in *Rosa26-Lmo2 + Aid-Cre* and *Rosa26-Lmo2 + Mb1-Cre* mice. PCR analysis of TCR gene rearrangements in infiltrated thymuses of diseased *Rosa26-Lmo2 + Aid-Cre* and *Rosa26-Lmo2 + Mb1-Cre* leukemic mice. Sorted DP T cells from the thymus of healthy mice served as a control for polyclonal TCR rearrangements. Leukemic thymus shows an increased clonality within their TCR repertoire (indicated by the code number of each mouse analyzed).

F   BCR clonality in *Rosa26-Lmo2 + Aid-Cre* and *Rosa26-Lmo2 + Mb1-Cre* mice. PCR analysis of BCR gene rearrangements in infiltrated thymuses of diseased *Rosa26-Lmo2 + Aid-Cre* and *Rosa26-Lmo2 + Mb1-Cre* leukemic mice. Sorted CD19[+] B cells (B cells) from spleens of healthy mice serve as a control for polyclonal BCR rearrangements. DP T cells from the thymus of healthy mice served as a negative control. Leukemic thymus shows an increased clonality within their BCR repertoire (indicated by the code number of each mouse analyzed).

behavior, leading to the characteristic clinical features of full-blown T-ALL. Therefore, this traditional model considers that the phenotype of the tumor cells mirrors that of the normal T cell that originally gave rise to the tumor. However, there is another possible way of interpreting the specific relationship between *LMO2* and T-ALL, and it is to consider that the *LMO2* oncogene is directly

capable of imposing the phenotypic characteristics of the tumor in a non-T target cell. In fact, *Rag-1* expression has been detected in early progenitors in both mice and humans (Boiers *et al*, 2013, 2018), therefore providing a mechanistic possibility for translocations to happen at very early hematopoietic developmental stages. If this second option is true, and LMO2 can indeed impose a T-cell

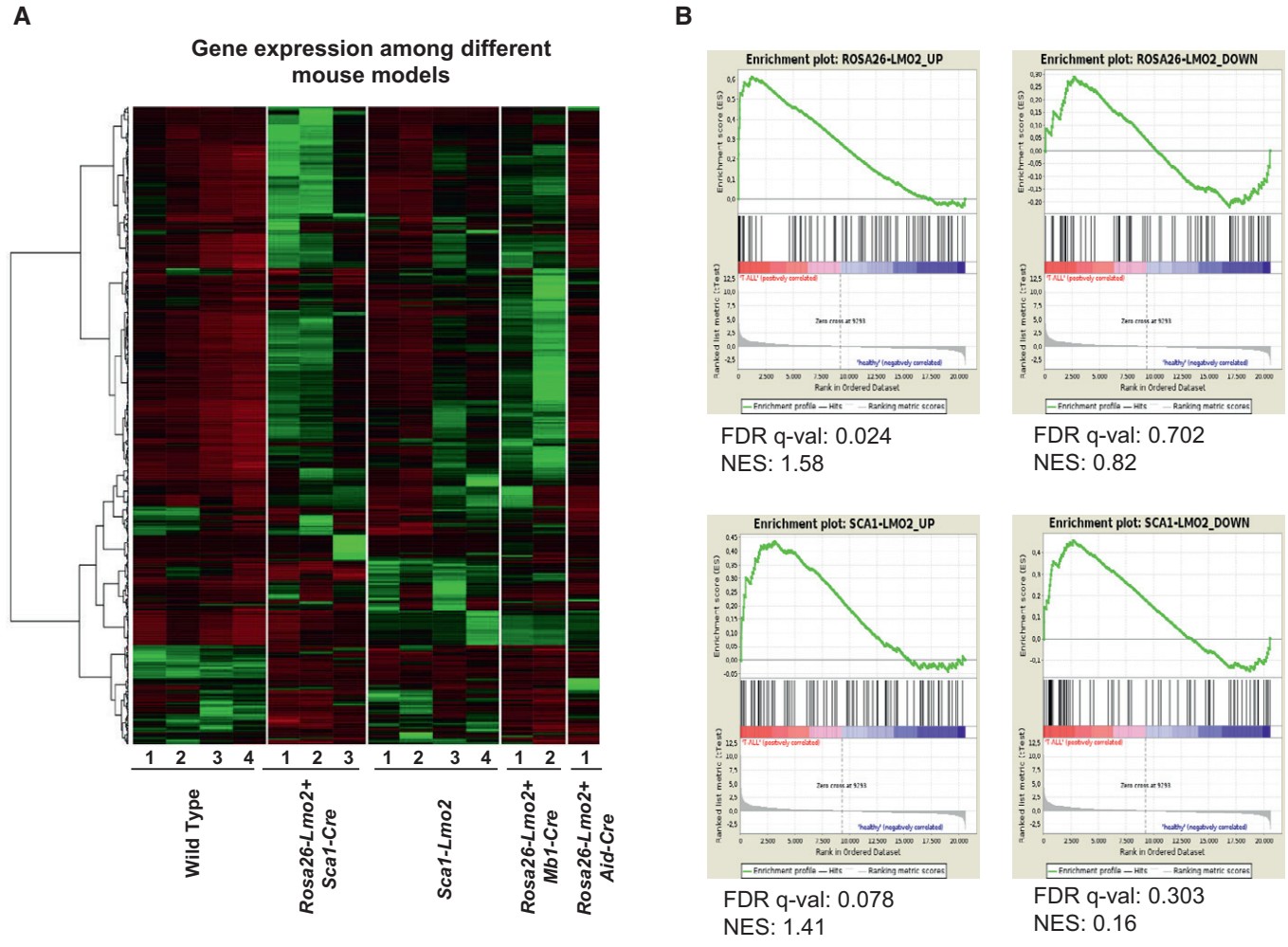

**Figure 6. Comparison of RNA-seq data from depicted mouse models compared to a human cohort of T-ALL.**

A Gene expression of *Rosa26-Lmo2 + Sca1-Cre*, *Sca1-Lmo2*, *Rosa26-Lmo2 + Mb1-Cre*, and *Rosa26-Lmo2 + Aid-Cre* with WT thymus as comparison. The 500 genes with the highest variance among the murine groups were chosen, and their corresponding FPKM values transformed to standard scores for visualization. [Row clustering was conducted with the ward.D method.]

B Gene set enrichment analysis (GSEA) of *Rosa26-Lmo2 + Sca1-Cre* and *Sca1-Lmo2* mouse-based gene signatures, against a human T-ALL childhood expression set with healthy controls. Mouse-based signatures consist of the 100 most up- and downregulated human homologue genes, as identified in a differential gene expression analysis between *Rosa26-Lmo2 + Sca1-Cre* versus WT and *Sca1-Lmo2* versus WT, respectively.

program in a non-T target cell, it would be difficult, however, to prove this fact in human tumors, since the deconvolution of the sequential events in the evolution of the leukemia becomes almost impossible due to the clonal and sub-clonal accumulation of genetic alterations by the time of the clinical presentation of the full-blown T-ALL. This clonality implies that in human T-ALL, in spite of its cellular heterogeneity, all leukemic cells carry the same *LMO2* initiating oncogenic genetic lesions, and this would seem to suggest a homogenous mode of action for *LMO2* within all cancer cells. However, there are other findings strongly pointing toward a reprogramming effect of non-T-cell lineage cells by LMO2. First, *LMO2* ectopic activation caused by retroviral insertion in the CD34[+] HSCs of X-SCID patients specifically triggered T-ALL development, but no other hematopoietic tumors (Hacein-Bey-Abina *et al*, 2008; Howe *et al*, 2008), although it is considered that *LMO2* expression in BM

progenitors is not relevant *per se* (Ruggero *et al*, 2016). And second, *Lmo2* expression in murine blood cells cooperates in the generation of iPS cells (Batta *et al*, 2014; Riddell *et al*, 2014). However, in order to prove that, for T-ALL development, *LMO2* expression does not need to be maintained beyond the initial step of reprogramming, one would require an experimental system capable of limiting the expression of *LMO2* to the target cell-of-origin compartment, since otherwise it would be impossible to discard a function for LMO2 in posterior tumor development, as exemplified by the *Rosa26-Lmo2 + Sca1-Cre* model. Such a "cell-of-origin-restricted" system would therefore allow us to prove, if these was indeed the case, that the oncogenes, like *LMO2*, capable of initiating T-ALL formation, might however be dispensable for posterior tumor progression and/or maintenance. In this study, we provide experimental evidence illustrating that HSC/PC and B cells

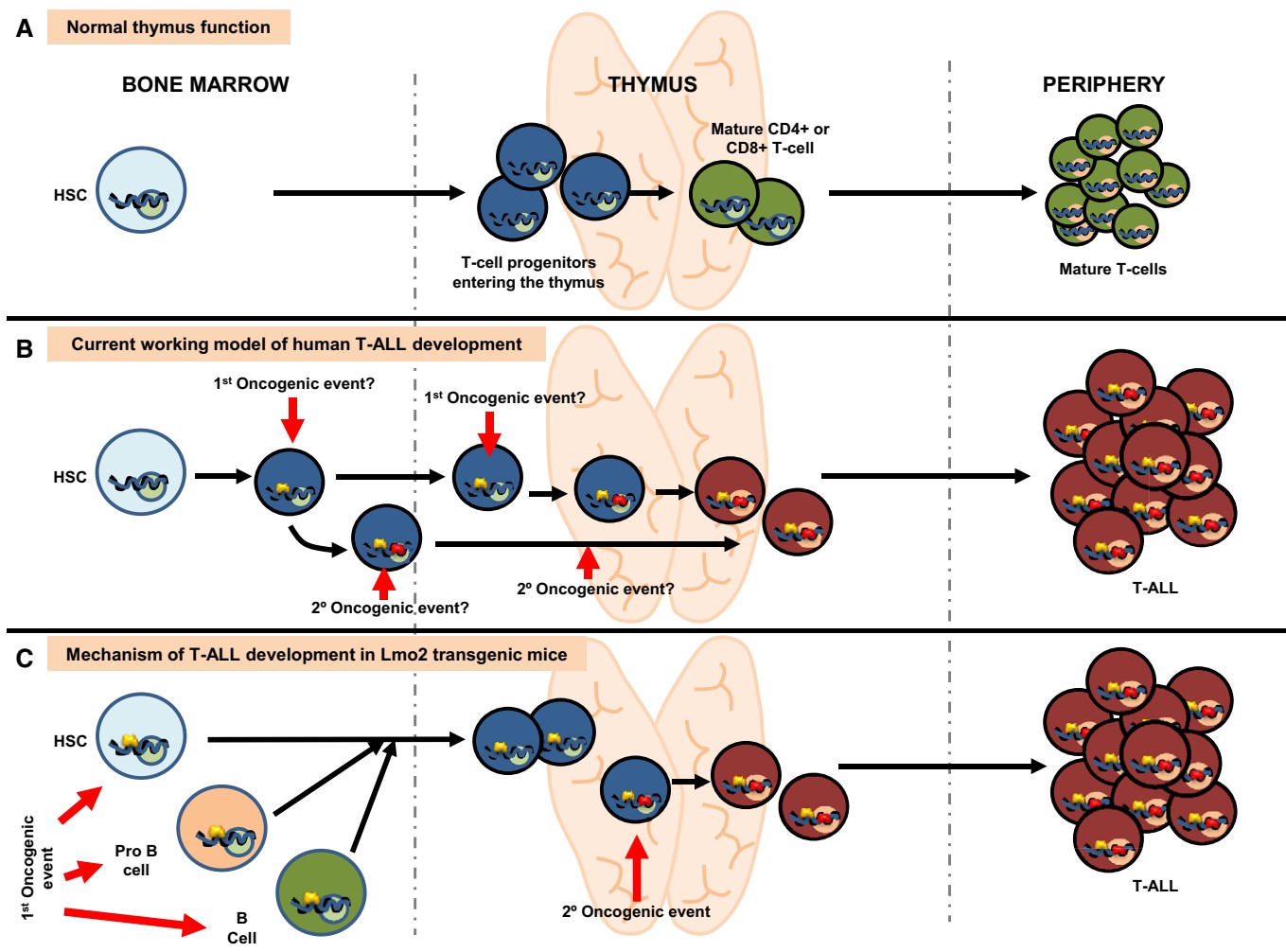

**Figure 7. A model by which ectopic expression of *Lmo2* reprograms HS/PCs and B cells into tumor T cells.**

A   Normal lymphoid development in human and mice. Blue circles represent normal gene regulatory events (activating or repressing) happening during T lymphocyte development. Green circles represent normal gene regulation events happening during terminal differentiation.

B   Current working model for the development of T-ALL in humans. The existence of dormant alterations previous to the terminal differentiation is unknown. The nature of both the cancer cell-of-origin and the cellular place where the second hit is taking place is therefore unknown.

C   Mechanism of T-ALL development in *Sca1-Lmo2* transgenic mice. Open yellow circles represent latent epigenetic regulatory events caused by Sca1-driven expression of *Lmo2*. These epigenetic marks do not interfere with normal T-cell development but become active (either activating or repressing) in the process of terminal differentiation when the second hit appears within the thymus, thus leading to the appearance of tumor T cells. According to this model, tumor T cell is the result of a cell reprogramming process that can be initiated even in committed B cells (see text for details).

are uniquely sensitive to transformation by *Lmo2* oncogene. However, within the hematopoietic system, not all cells are equally permissive to transformation. Restricted Cre-mediated activation of *Lmo2* in different stages of B-cell development induced T-ALL. However, the differentiation state of the B cell-of-origin influences the latency, but it provides thoroughly and unexpectedly a T-cell phenotype. This is a novel phenomenon in contrast to the previous assumptions; i.e., the phenotype of the leukemia cells is identical to the cell-of-origin (Vicente-Duenas *et al*, 2013). These results indicate that the T-ALL cell-of-origin must possess sufficient plasticity to allow the tumoral reprogramming to take place or, at least, to be initiated. Thus, Lmo2 has the power and capacity to switch from a B-cell fate to a T-cell neoplasia, although Lmo2 does not seem to contribute to the generation of normal T lymphocytes. This finding contrast with the role play by Pax5, whose deletion of this master regulator of the B-cell lineage reprograms B cells into functional T lymphocytes which only occasionally gives rise to T-cell malignancies (Rolink *et al*, 1999; Cobaleda *et al*, 2007). This mechanism of Lmo2-dependent reprogramming has been reported in other contexts outside of malignancy, like *Lmo2* in iHSCs (Riddell *et al*, 2014). Thus, the data presented here suggest a more general role for LMO2 to shape the epigenome or to be involved in chromatin remodeling early on in T-ALL disease and it would not be surprising that other important drivers for human T-ALL, like *SCL*, *LMO1*, or HOX11/TLX1, contribute to the neoplasm through a similar reprogramming mechanism.

The multiple genetic hits required for T-ALL development can be related to the fact that the changes, which are necessary for reprogramming mature cells to pluripotent phenotype, are inherently disfavored developmentally. In this case, biological barriers try to prevent cells from changing their identity in order to avoid the risk of malignant transformation. Evidence to support the inherent resistance of cells to reprogramming by an oncogene to a tumor phenotype comes from recent studies of stem-cell-based animal models of human cancer. For instance, in a stem-cell-based transgenic model of multiple myeloma, the loss of *p53* accelerated the appearance of disease by allowing the *MafB* oncogene to drive a much more efficient malignant reprogramming (Vicente-Duenas *et al*, 2012b,c). Something similar happens in the case of mucosa-associated lymphoid tissue (MALT) lymphoma that is driven by the *MALT1* oncogene (Vicente-Duenas *et al*, 2012a). In a stem-cell-based model of CML, restoration of p53 activity slowed the progression of the disease and extended the survival of leukemic animals by inducing the apoptotic death of primitive leukemic cells (Velasco-Hernandez *et al*, 2013). Similarly, a significant proportion of T-ALL in all our murine models carried *p53* loss-of-function mutations (Table EV4) facilitating pathological reprogramming to a malignant T-cell phenotype.

We propose that in human T-ALL, genetic alterations of *LMO2* may act in a hit-and-run fashion in early precursors, while evolved tumor cells are reliant on alternative oncogenic mechanisms. The presence of Lmo2 is necessary for the early stages of transformation, but the final tumor phenotype is determined by the niche. In the present results, the final phenotype of the T-ALL is defined by the thymus environment (Fig 7). This may provide an explanation for the failure of some modern targeted therapies to clear tumor stem cells, despite being effective agents against evolved tumor cells. As a consequence, treatment strategies targeting oncogenic pathways that are active in both the early and late stages of tumor development may be needed to eradicate completely T-ALL. These findings on the mechanisms of cellular commitment to a tumoral fate by LMO2, therefore, have important implications for understanding and therapeutically targeting T-ALL tumor cells and to regenerative medicine, since it will be essential to have full control over the potential malignancy of reprogrammed cells.

# Materials and Methods

Detailed methods can be found in the Appendix Supplementary Methods available online.

### Generation of mouse strains

All animal work was conducted in accordance with national and international guidelines on animal care and was approved by the Bioethics Committee of the University of Salamanca and the Bioethics Subcommittee of Consejo Superior de Investigaciones Cientificas (CSIC). The *Rosa26-Lmo2* mice were bred to *Sca1-Cre* (Mainardi *et al*, 2014), *Mb1-Cre* (Hobeika *et al*, 2006), or *Aid-Cre* mice (Crouch *et al*, 2007) to generate *Rosa26-Lmo2 + Sca1-Cre*, *Rosa26-Lmo2 + Mb1-Cre*, and *Rosa26-Lmo2 + Aid-Cre* mice, respectively. The *Sca1-Lmo2* vector was generated by inserting the *TdTomato–IRES*-mouse *Lmo2* cassette into the ClaI site of the pLy6E

vector (Miles *et al*, 1997), and the transgene (2 ng/µl) was injected into CBAxC57BL/6J fertilized eggs.

Upon clinical manifestations of disease, mice were sacrificed and subjected to standard necropsy procedures. All major organs were examined under the dissecting microscope. Tissue samples were taken from homogenous portions of the resected organ and fixed immediately after excision. Differences in survival of transgenic and control *WT* mice were analyzed using the log-rank (Mantel–Cox) test.

### Array-comparative genomic hybridization (aCGH)

Whole-genome analysis was conducted using *Mus musculus* whole-genome 4x180k oligonucleotide aCGH (AMADID 27411; Agilent Technologies) following the standard protocol. Copy number-altered regions were detected using ADM-2 (set as 6) statistics with a minimum number of five consecutive probes.

### Gene expression microarray analysis of murine tumors

Tumoral and normal thymuses were harvested from 10 *Sca1-Lmo2* mice and 4 control littermate WT mice, respectively. Samples were analyzed using Affymetrix Mouse Gene 1.0 ST arrays. A cutoff of FDR < 0.05 was used for the differential expression calculations. All analyses were performed using R and Bioconductor software.

### Mouse and human exome library preparation and next-generation sequencing

Mouse exome library preparation was performed using the Agilent SureSelectXT Mouse All Exon kit with modifications. Furthermore, 2 × 100 bp sequencing with a 6-bp index read was performed using the TruSeq SBS Kit v3 on the HiSeq 2500 (Illumina). Fastq files were generated using BcltoFastq 1.8.4 (Illumina). BWA version 0.7.4 was used to align sequence data to the mouse reference genome (GRCm38.71). Human translocation t(11;14)(q13;q11) bone marrow samples were obtained after informed consent within the AIEOP-BFM- ALL study and processed via the Macrogen Europe platform, Heidelberg.

### RNA sequencing

RNA sequencing libraries were generated from 500 ng of total RNA using the TruSeq RNA sample prep kit (Illumina) from the blast cells obtained different mouse models employed in the study, including cells from healthy thymus as a control (wt). Later, the libraries were subjected to 2 × 100 bp paired-end sequencing using HiSeq 2000 instrument (Illumina). The RNA-seq data were aligned against the mouse reference genome (mm10/GRCm38.83) with TopHat 2.1.0 (Kim *et al*, 2013), and FPKM values per gene were calculated with the R/Bioconductor package bamsignals (Mammana & Helmuth, 2016). Genes with FPKM values > 1 in less than three samples were excluded from the analysis. We selected the 500 genes with the highest variance over all samples, transformed the corresponding FPKM values into standard scores (Table EV5), and visualized the results with the R/Bioconductor package gplots' heatmap.2 function (row dendrogram, clustering method "ward.D"; Warnes *et al*, 2016).

Differential analysis between the mouse groups *Rosa26-Lmo2 + Sca1-Cre* and *Sca1-Lmo2* against the wild-type group was conducted with DESeq2 (Love *et al*, 2014), with a minimum adjusted *P*-value of 0.05. Signatures of the 100 most significantly up- and downregulated human homologue genes per differential analysis were then tested for enrichment with the Broad Institute's GSEA tool (Table EV6; Mootha *et al*, 2003; Subramanian *et al*, 2005), against a human childhood T-ALL set with corresponding wild-type thymus control samples (Ng *et al*, 2014).

### Statistical analysis

Differences between the transgenic and control littermate wild-type mice in the percentage and an absolute number of thymocytes were analyzed by analysis of variance (ANOVA) followed by the Kruskal–Wallis and Dunn's multiple comparison tests. Differences in survival of transgenic and control WT mice were analyzed using the log-rank (Mantel–Cox) test. *P*-values < 0.05 were considered statistically significant. Statistical analysis and data representation were performed using the GraphPad Prism 5.00 software (San Diego, California, USA).

### Accession numbers

The mouse RNA-seq and gene expression microarray data have been deposited in NCBI's Gene Expression Omnibus (Edgar *et al*, 2002) and are accessible through GEO Series accession number GSE83572.

**Expanded View** for this article is available online.

### Acknowledgements

We are indebted to all members of our groups for useful discussions and for their critical reading of the manuscript. J.H. has been supported by the German Cancer Aid (Project 110997 and Translational Oncology Program 70112951), the German Jose Carreras Leukemia Foundation (DJCLS 02R/2016), Deutsches Konsortium für Translationale Krebsforschung (DKTK), Joint funding (Targeting MYC L*10), the Kinderkrebsstiftung (2016/17), and the "Elterninitiative Kinderkrebsklinik e.V. Düsseldorf". SG has been supported by a scholarship of the Hochschule Bonn-Rhein-Sieg. AB has been supported by the German Children's Cancer Foundation and the Federal Ministry of Education and Research, Bonn, Germany. Research in ISG group is partially supported by FEDER and by MINECO (SAF2012-32810, SAF2015-64420-R, and Red de Excelencia Consolider OncoBIO SAF2014-57791-REDC), Instituto de Salud Carlos III (PIE14/00066), ISCIII- Plan de Ayudas IBSAL 2015 Proyectos Integrados (IBY15/00003), by Junta de Castilla y León (BIO/SA51/15, CSI001U14, UIC-017, and CSI001U16), Fundacion Inocente Inocente, and by the ARIMMORA project (European Union's Seventh Framework Programme (FP7/2007-2013) under grant agreement no. 282891). ISG Lab is a member of the EuroSyStem and the DECIDE Network funded by the European Union under the FP7 program. AB and ISG have been supported by the German Carreras Foundation (DJCLS R13/26). IGR was supported by BES-Ministerio de Economía y Competitividad (BES-2013-063789). AML and GRH were supported by FSE-Conserjería de Educación de la Junta de Castilla y León (CSI001-13, CSI001-15). Research in CVD group is partially supported by FEDER, "Miguel Servet" Grant (CP14/00082—AES 2013-2016) from the Instituto de Salud Carlos III (Ministerio de Economía y Competitividad), "Fondo de Investigaciones Sanitarias/Instituto de Salud Carlos III" (PI17/00167), and

by the Lady Tata International Award for Research in Leukaemia 2016–2017.

### Author contributions

Conception and design of the project: IG-R, CV-D, AB, IS-G, and JH; development of methodology: IG-R, SB, SGT-D, AM-L, GR-H, IG-H, SP, YN, AO, VD, BP, OB, DA-L, JDLR, RJ, MBGC, FJGC, and CV-D; data acquisition: IG-R, SB, CW, AM-L, GR-H, IG-H, SP, YN, AO, VD, BP, OB, DA-L, JDLR, RJ, MBGC, FJGC, CV-D, ISL, and IS-G; management of patient samples: SB, MD, MSt, MSc, WW, OH, AB, and JH; analysis and interpretation of data (e.g., statistical analysis, biostatistics, computational analysis): IG-R, SB, AM-L, GR-H, IG-H, CW, SP, YN, AO, VD, BP, OB, DA-L, JDLR, RJ, MBGC, FJGC, CV-D, ISL, IS-G, AB, and JH; writing, review, and/or revision of the manuscript: IG-R, SGT-D, AM-L, GR-H, IG-H, CW, SP, YN, AO, VD, BP, OB, DA-L, JDLR, RJ, MBGC, FJGC, CV-D, ISL, IS-G, AB, and JH; administrative, technical, or material support (i.e., reporting or organizing data, constructing databases): IGS, IG-H, YN, DA-L, IS-G, SB, JH, and AB; and study supervision IG-R, CV-D, IS-G, JH, and AB.

### Conflict of interest

The authors declare that they have no conflict of interest.

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
