## [Review Process File · The EMBO Journal]

Lmo2 expression defines tumor cell identity during T-cell leukemogenesis

Idoia García-Ramírez, Sanil Bhatia, Guillermo Rodríguez-Hernández, Inés González-Herrero, Carolin Walter, Sara González de Tena-Dávila, Salma Parvin, Oskar Haas, Wilhelm Woessmann, Martin Stanulla, Martin Schrappe, Martin Dugas, Yasodha Natkunam, Alberto Orfao, Verónica Domínguez, Belén Pintado, Oscar Blanco, Diego Alonso-López, Javier De Las Rivas, Alberto Martín-Lorenzo, Rafael Jiménez, Francisco Javier García Criado, María Begoña García Cenador, Izidore S. Lossos, Carolina Vicente-Dueñas, Arndt Borkhardt, Julia Hauer, and Isidro Sánchez-García.

Review timeline:

Submission date:	8 th December 2017
Editorial Decision:	12 th January 2018
Revision received:	9 th April 2018
Editorial Decision:	24 th April 2018
Revision received:	29 th April 2018
Accepted:	1 st May 2018

Editor: Daniel Klimmeck

Transaction Report:

1st Editorial Decision

12th January 2018

Thank you for the submission of your manuscript (EMBOJ-2017-98783) to The EMBO Journal. Your study has been sent to two referees, and we have received reports from both of them, which I copy below.

As you will see, the referees acknowledge the potential high interest of your work, although they also express a number of major issues that will have to be addressed before they can support publication of your manuscript in The EMBO Journal. In more detail, referee #1 is concerned about lack of novelty of the first part of your study, as to partial overlap of your findings on aberrant Lmo2 with earlier studies (ref#1, standfirst, pt. 3). Further, this referee states that your claims regarding relevance for human T-ALL are not sufficiently well supported by the current data (ref#1, pt. 2). Referee #2 agrees in that the human link of the results is not sufficiently made by the microarray-based analysis and asks you to corroborate these aspects by RNAseq (ref#2, pts. 1,4). This referee is also points to lack of proof for absence or presence Lmo2 at protein level (ref#2, pt.2). In addition, the referees list a number of issues related to technical documentation, FACS data illustration and missing controls, which need to be addressed to achieve the level of robustness required for The EMBO Journal.

I judge the comments of the referees to be generally reasonable and we are in principle happy to invite you to revise your manuscript experimentally to address the referees' comments. Please note however, that we would need strong support from the referees on such a revised version of the manuscript to move towards publication. I agree that it would be essential to revise and condense the confirmatory aspect in the first part of the study, and expand on the novel aspects related to B-cell-driven T-ALL.

REFeree REPORTS.

Referee #1:

In this study, García-Ramírez et al. study the cell of origin of LMO2 induced leukemias, by exploiting a novel conditional LMO2 knockin mouse model. To study the role of LMO2 in HSCs, the authors cross the model to Sca1-Cre. These experiments show that aberrant LMO2 expression in HSCs/PCs, create a preleukemic state and eventually induce T-ALL. The significance of these findings are limited, given that similar findings have been previously reported in literature. Next, the authors used another sca1-LMO2-TdTomato transgenic model to show that Lmo2 expression is required to induce a pre-leukemic state within the thymus but is dispensable for clonal T-ALL transformation. By crossing Sca1-Lmo2 mice with immunodeficient nude (nu/nu) mice, the authors demonstrate that a proficient thymus is required for T-ALL development in this model. Finally and unexpectedly, the author finally cross their condition knockin model with B lineage are lines such as Mb1-Cre and AID-Cre, and show that some of these animals also develop T-ALL. Altogether, this is an interesting and provocative study on the role of LMO2 in leukemia development. The novelty mainly resides in the second part of the manuscript.

Comments

1. The authors show in their Sca1-LMO2-TdTomato model that the ultimate T-ALL tumors are TdTomato negative. I think that these data might suggest that LMO2 would first induce a preleukemic state in the double negative thymocytes in their model. However, these preleukemic cells have not been fully arrested and subsequent hits might push them further into differentiation before they eventually undergo full leukaemia transformation. In that case, the fact that these leukemias are TdTomato negative might just reflect the fact that the eventual cell of origin of the fully transformed T-ALL is an LMO2 negative cell. I'm not sure if the authors would agree with this scenario based on their interpretation as shown in this manuscript.

2. Along the same lines and importantly, one should also consider that in the human situation, the LMO2 expression will not disappear in a fully transformed cell, because LMO2 expression is driven by TCR enhancer in the t(11;14). Therefore, the relevance of these data for the human disease is questionable. Along the same lines, the authors state in their discussion: "We propose that in human T-ALL, genetic alterations of LMO2 may act in a hit- and-run fashion in early precursors, while evolved tumor cells are reliant on alternative oncogenic mechanisms. The presence of Lmo2 is necessary for the early stages of transformation but the final tumor phenotype is determined by the niche. In the present results, the final phenotype of the T-ALL is defined by the thymus environment (Figure 6)." However, and as mentioned above, LMO2 in human T-ALL is mainly activated by T cell receptor driven translocations. Therefore, LMO2 will only become activated in a cell stage in which the TCR loci are undergoing rearrangements. I believe that LMO2 activation in a Sca1 positive cell does not reflect this scenario. To really mimic the human disease, LMO2 should be activated by a later T-cell specific Cre line, that becomes active at the same time as the TCR-LMO2 would occur in human T-cell precursor. This point should be better addressed and discussed throughout the manuscript.

3. As stated by the authors on page 8 in the results section: "The majority of Sca1-Lmo2 T-ALL cases (n=21) were TdTomato-." So I guess this means that some of the tumors were still TdTomato positive? If so, this mimics what has previously been reported for the CD2-LMO2 mouse model (Rabbits, McCormack and Utpal Dave lab), namely that LMO2 activation in early precursors can induce 2 types of murine T-ALL, i.e.. early immature T-ALL (hhex, Mycn and Iy11 positive; these would be still TdTomato positive in this model) as well as mature murine T-ALLs (these would be TdTomato negative in this model). It might be interesting to also look at the TdTomato positive tumors in this model and see if the pattern of mutations, and the expression profile indeed is different.

3. Unexpectedly, the authors show that LMO2 activation in B-lineage also results in T-ALL (lower frequency). My main concern with these experiments would be potential leakage of these Cre Lines

in early hematopoietic precursor cells or in the T-Lineage. Can the authors rule out that there might be any leakage associated with these lines. For example, with the AID-Cre line, it has been shown that this might be the case.

4. Throughout the manuscript, the FACS gates are not kept consistent in the comparison between WT and TG animals. For example, Suppl Fig 2. Please correct throughout the manuscript.

5. Suppl Fig1C: A double peak is visible for GFP. Please explain

6. Fig5C + Supp Fig 5C: How do the authors explain that in the preleukemic situation, there are mature CD4⁺ as well as mature CD8⁺ T cells that are positive for TdTomato? I guess the scd1 promoter is not functional in these cells? Please explain.

7. Figure 3A. What cells are used as control population for the comparison with the signature from the tumors? It is important to use the appropriate normal T cell control in these experiments.

8. What type of tumours were generated in nu/nu mice? What was their immunophenotype?

Referee #2:

T-ALL accounts for 15% of all childhood acute leukaemia cases. Here, TAL1 complexes with E2A/HEB binding subsequently to RUNX3, LMO1/2 and GATA3 forming a transcriptional scaffold complex that binds to the intergenic/intronic enhancer regions. The first two hits in T-ALL are supposedly TAL1 and LMO1/2 activation by different mutational events forming a chromatin regulatory circuit or looping that drives oncogene expression, which is well described in the literature. The third hit is usually hyperactivation of NOTCH signalling driving oncogenic MYC amplification that further invades the chromatin regulatory circuit boosting further reprogramming of malignant T-cells. The cell of origin of T-ALL was also described to be an immature thymocyte that originates from different stages of blocked early thymic development, where both the cortical as well as the subcapsular zone of the thymus for T-cell developmental stages were shown to be involved. How frequent a committed B-cell retro-differentiation program in human T-ALL would be relevant remains enigmatic. Still, this transgenic mouse model work is an elegant illumination of a key role of LMO2 as a genetic gardener to shape the transcriptome and even somatic mutation landscape of T-ALL. It is known since a long time that LMO2 is one of the most frequently mutated genes in T-ALL and the authors describe it here with a transient expression by CRE-mediated recombination to be a main transcriptional reprogramming transcription factor that initiates a unique transcriptome of T-ALL. It even has the power and capacity similar like Pax5 loss to switch from a B-cell fate to a T-cell neoplasia, here the authors could discuss that part of the work better in light of Dr.'s Meinrad Busslinger and Stephen Nutt works which according to the reviewer opinion is a rare but another such interesting example for T-cell neoplasia etiology. Similar findings were e.g. also described in colorectal cancer by retro-differentiation by the Dr. Gretchen lab (Schwitalla et al., Cell, 2013) of epithelial differentiation being reprogrammed to intestinal stem cells by hyperactive cytokine and WNT signalling. Thus, the paper overall is well carried out, has a wealth of model work and genetic analysis, where only one major point and a few minor points could be performed to improve the impact of the study to the field of T-ALL.

Major:

1) A true strength of the study is the somatic mutation and gene copy number gain or deletion analysis and that four different models for T-ALL were developed. However, a conclusion from the microarray analysis to similarity with human T-ALL is not justified. Better and more state-of-the-art is for sure an RNA-seq expression profile where e.g. the most significantly 500 genes from human T-ALL up- or down-regulated are compared to the murine T-ALL models used in this study. This heat map analysis would justify better for a conclusion. Thus, RNA-seq analysis should be performed on 3-4 controls, 3-4 T-ALL samples of genetic model 1 (Figure 1) and 3-4 T-ALL samples of genetic model 2 (Figure 2). If the authors can, one of the two B-cell identity driven third models for T-ALL developed in this study could also be profiled by RNA-seq to allow for a better conclusion if that also matches closely the transcriptome of human T-ALL or not and how well comparable it might be to their other more solid genetic models due to penetrance and latency. The

nude mouse model work in Figure 4 is mechanistically interesting, but probably would not mimic closely a situation in patients with the exception of rare cases of SCID phenotype patients developing T-ALL in association with gene therapy as introduced by the authors and RNA-seq profiling here is not critical. Overall, a comparative RNA-seq analysis would be a true reality check with human T-ALL and a side-by-side comparison with conclusion of their best established model is a key finding. This could be a very strong Figure and conclusion for the work performed is then more transparent to the reader. Unfortunately as the data were recorded, current analysis cannot be judged which model is better or closest to human T-ALL, which would be advisable to be incorporated by simple RNA-seq analysis in this study. An alternative could be to compare their Affymetrix data to human Affymetrix data in similar reasoning as outlined above.

2) One important aspect and weakness of the paper is the lack of proof for presence or absence of LMO2. It would need to be a true controlled proof both for mRNA and protein data for LMO2 loss after T-ALL phenotypes have been established to allow for justification of the claim. The tomato-reporter data and immunohistochemistry in Figure 2 are not conclusive. Thus, the author should provide from sorted neoplastic cell populations of T-ALL where they make the claim that LMO2 is indeed lost with appropriate controls real-time mRNA expression quantification and direct Western blotting for LMO2 protein. This is particularly concerning where the authors make microarrays and write "...Cells were not sorted prior to RNA extraction for this analysis...." But cells are heterogeneous/mosaic from the mice with and without LMO2 expression as clearly evidenced by reporter gene expression. Here, the authors need to give more detailed methods and re-write the results based on which cell populations were profiled. Certainly, the major purified neoplastic cell population is of most interest here and data could be viewed "contaminated" by unwanted cell types. The claim of LMO2 presence or absence needs then to be matched with sorted populations and a purity check on neoplastic cells that were sorted is mandatory to control for the procedure and to characterize the genetics of the models which is diverse and from text result body description classified as homogenous.

3) The gating performed in the FACS plots is often not controlled, the gates are touching the X- and/or Y-axis, which should never be the case, FACS data have to be reanalysed in improved and better controlled way, where e.g. a wildtype thymus control has four gates for DN and DP to CD4 or CD8 single positive stages (one can e.g. look at Figure 1/2 on the gating of wt thymus compared to the experimental mice thymi) then the exact same gating shall be applied throughout the manuscript.

4) Here, the authors use six tables to emphasize on their genetic findings, but a key strength would be to link similarities to human T-ALL simply by referencing. Thus, the authors could reduce the number of Tables to a more comprehensive format and add where possible human mutational or genetic gene loss or amplification literature findings. This would allow for a better description of similar findings in human T-ALL. Such a comprehensive Table can facilitate the understanding of the reader that model x is most similar to human T-ALL findings with genetic key findings as established here in their paper. Many cross-references are already in the text body and if reference limitations are an issue then all references for that comprehensive table could be also moved to a place in the paper with assistance of the editorial team.

Minor:

- 1) The abstract contains unnecessary overstatements right from start and should be improved. Several wrong assumptions are given. T-ALL is from cancer genome landscape and cell fate decisions pretty well described and understood. See e.g. reviews by A. T. Look or S. Armstrong dating back to 2005 or newer ones.
- 2) To description of Appendix Fig. S1A-B: it should be clarified that In the absence of Cre, neither Lmo2 nor eGFP are expressed, due to a STOP cassette in front of Lmo2 that has to be removed by CRE recombinase action to allow for their expression.
- 3) Certainly, it should be critically discussed that providing foreign transgenes to mice will raise unwanted immunogenicity in case of fluorescent protein expression or reporter usage that will be an issue.

4) The CRE used in a subset of studies is a Sca1-Cre and Sca1 is a downstream target gene of STAT3 action, which could be cited. This would suggest that at least also more committed myeloid or lymphoid progenitor cell populations and not "stem cell" restricted cells recombine, since cytokine-driven STAT3 was described to drive SCA1 in mice also in more committed or differentiated cell types and not as suggested only in stem cells or HSC. This would be too vague or an overstatement, if at all then a restricted hematopoietic progenitor expression due to Sca1-CRE action might be concluded.

5) The discussion could benefit from a few more points that could be relevant to bring the authors conclusions: Here, main focus was given on the potential of LMO2 to reprogram the T-cell lineage and it was described to be one of six key factors able to initiate induced HSC reprogramming, being also involved in B-cell neoplasias. The data suggest a more general role for LMO2 to shape the epigenome or to be involved in chromatin remodelling early on in T-ALL disease. It is not clear for the reviewer if in human T-ALL activation of LMO2 has to be switched off or if lower expression of LMO2 can be found in T-ALL. Also the second family member LMO1 might be similar in that regard to LMO2 or not, which should be discussed better.

6) Abbreviations should be defined at first use, examples are DLBCL or BCP-ALL, etc.

7) Fig. 1B Western is of poor quality and should be replaced.

8) Statistics are incomplete in Appendix Fig. S1E, since the diseased cells were not incorporated, if it is not significant that should be written, but it looks not to be analysed.

9) The sentence in page 12 makes little to no sense where it says:.....Malignant T-cells were primarily CD8+/-CD4+/-.....at the end it is not the reader who can gamble what kind of cells the authors might refer too, in the Figure they claim the diseased cells are all double positive for CD4/CD8 and that could be a wrong description. Figure 5B has also single positive cells if one really looks at brightness of the cells.

1st Revision - authors' response

9th April 2018

Point-by-point response to the referees' comments

Referee #1:

In this study, García-Ramírez et al. study the cell of origin of LMO2 induced leukemias, by exploiting a novel conditional LMO2 knockin mouse model. To study the role of LMO2 in HSCs, the authors cross the model to Sca1-Cre. These experiments show that aberrant LMO2 expression in HSCs/PCs, create a preleukemic state and eventually induce T-ALL. The significance of these findings are limited, given that similar findings have been previously reported in literature. Next, the authors used another sca1-LMO2-TdTomato transgenic model to show that Lmo2 expression is required to induce a pre-leukemic state within the thymus but is dispensable for clonal T-ALL transformation. By crossing Sca1-Lmo2 mice with immunodeficient nude (nu/nu) mice, the authors demonstrate that a proficient thymus is required for T-ALL development in this model. Finally and unexpectedly, the author finally cross their condition knockin model with B lineage are lines such as Mb1-Cre and AID-Cre, and show that some of these animals also develop T-ALL. Altogether, this is an interesting and provocative study on the role of LMO2 in leukemia development. The novelty mainly resides in the second part of the manuscript.

We would thank to reviewer 1 for his/her thorough review of our manuscript and his/her very positive comments and suggestions for improving our manuscript. Referee 1 indicates, quite rightly, some weak points. Now we have carefully addressed all comments in the revised manuscript as detailed below:

Comments

1. *The authors show in their Sca1-LMO2-TdTomato model that the ultimate T-ALL tumors are TdTomato negative. I think that these data might suggest that LMO2 would first induce a preleukemic state in the double negative thymocytes in their model. However, these preleukemic cells have not been fully arrested and subsequent hits might push them further into differentiation before they eventually undergo full leukaemia transformation. In that case, the fact that these leukemias are TdTomato negative might just reflect the fact that the eventual cell of origin of the fully transformed T-ALL is an LMO2 negative cell. I'm not sure if the authors would agree with this scenario based on their interpretation as shown in this manuscript.*

We also appreciated this possibility, but formally excluded it by observing both leukemia development in Sca1-Lmo2+nu/nu mice where tumor cells are Lmo2 positive cells and T-ALL development when we restrict Lmo2 expression to different stages of B-cell development.

2. *Along the same lines and importantly, one should also consider that in the human situation, the LMO2 expression will not disappear in a fully transformed cell, because LMO2 expression is driven by TCR enhancer in the t(11;14). Therefore, the relevance of these data for the human disease is questionable. Along the same lines, the authors state in their discussion: "We propose that in human T-ALL, genetic alterations of LMO2 may act in a hit- and-run fashion in early precursors, while evolved tumor cells are reliant on alternative oncogenic mechanisms. The presence of Lmo2 is necessary for the early stages of transformation but the final tumor phenotype is determined by the niche. In the present results, the final phenotype of the T-ALL is defined by the thymus environment (Figure 6)." However, and as mentioned above, LMO2 in human T-ALL is mainly activated by T cell receptor driven translocations. Therefore, LMO2 will only become activated in a cell stage in which the TCR loci are undergoing rearrangements. I believe that LMO2 activation in a Sca1 positive cell does not reflect this scenario. To really mimic the human disease, LMO2 should be activated by a later T-cell specific Cre line, that becomes active at the same time as the TCR-LMO2 would occur in human T-cell precursor. This point should be better addressed and discussed throughout the manuscript.*

The genotype-phenotype correlations established in humans have demanded during the last decades that we explain the nature of this intimate association between each genetic lesion (LMO2, in this case) and the phenotype with which it is associated (T-ALL for LMO2). Two different hypotheses have been considered to explain this link. In the classical view of the initiation and progression of T-ALL, the initiating genetic alteration takes place and is required for the immortalization of a committed/differentiated target cell. Such cell will afterwards acquire additional genetics hits over time. The acquisition of additional hits aggravates the deregulation of the behaviour of the differentiated target cell, therefore leading to

the clinically recognized features of T-ALL. This is the model that has traditionally been assumed in the study of *LMO2*, taking for granted that the phenotype of the tumor cells was a reflection of that of the normal cell that gave rise to the tumor in the first place. The other way of interpreting the genotype-phenotype correlations observed between *LMO2* and T-ALL is to consider the possibility that the oncogene is directly responsible of imposing the specific characteristics of the tumor phenotype. In this sense, recent studies have detected Rag-1 expressing progenitors much earlier in both mice and humans (Böiers C, et al., *Cell Stem Cell*. 2013 Nov 7;13(5):535-48, and Böiers C, et al., *Dev Cell*. 2018 Feb 5;44(3):362-377). However, deconvolution of the stepwise events taking place during tumor cell evolution is difficult because of the many genetic alterations that become clonally dominant by the time of their interrogation within the clinically manifested T-ALL, and the multitude of avenues by which any given tumor can evolve. As the reviewer indicates, in human T-ALL, all cancerous cells, with independence of the cellular heterogeneity existing within the T-ALL, carry the same *LMO2* initiating oncogenic genetic lesions. Overall these observations seem to point towards a homogenous mode of action for *LMO2* within cancer cells. However, despite frequent alterations of *LMO2* in hematologic tumors, its impact on lineage organization during leukemogenesis has remained unclear. Two sets of observations suggest a reprogramming effect of non-T-cell lineage cells by *LMO2*. First, *LMO2* expression due to retroviral insertion and transactivation in CD34+ HSCs of X-SCID patients caused T-ALL but no other hematopoietic tumors (*JCI*. 2008;118:3132-3142; *JCI*. 2008;118:3143-3150), although it is considered that *LMO2* expression in BM progenitors is not relevant per se (*Leukemia* 2016, 30: 1959-1962). And second, *Lmo2* expression in murine blood cells cooperates in the generation of iPS cells (*Cell*. 2014;157:549-564; *Cell reports* 2014;9:1871-1884). Indeed, in order to prove that the maintenance of the expression of the oncogene is not necessary for tumor progression beyond the initial step of reprogramming, one would need to find a way of restricting the expression of the oncogene to the stem/progenitor compartment, otherwise it will remain unmask as exemplified by the *Rosa26-Lmo2+Sca1-Cre* model. Such a system would also allow us to prove, if these was indeed the case, that the oncogenes that initiate T-ALL formation might be dispensable for tumor progression and/or maintenance. In this manuscript, we address this emerging and paradigm-altering area of research by using *in vivo* genetic lineage tracing to assess the capability of *LMO2* to reprogram HSCs. Therefore, we first initiated *Lmo2* expression in HSCs and maintained its expression in all hematopoietic cells. These mice develop exclusively aggressive human-like T-ALL. However, a definitive conclusion about an exclusive reprogramming effect of *Lmo2* in murine HSC/PC in contrast to its expression in T-cell precursors and mature T-cells was limited. Thus, we next modeled the scenario of HSC/PC restricted *Lmo2* expression *in vivo* showing that transient *Lmo2* expression in HSC/PCs is sufficient for oncogenic function and induction of human like T-ALL without the need for sustained *Lmo2* expression in the T-ALL bulk. ***Thus, we hoped to convey that our studies reveal a new dimension for how LMO2 might function as an oncogene in T-ALL, through a hit-and-run mechanism that has not been previously described. We have modified the discussion to better address this critique.***

3. As stated by the authors on page 8 in the results section: "The majority of *Sca1-Lmo2* T-ALL cases (n=21) were *TdTomato*-." So I guess this means that some of the tumors were still *TdTomato* positive? If so, this mimics what has previously been reported for the *CD2-LMO2* mouse model (Rabbits, McCormack and Utpal Dave lab), namely that *LMO2* activation in early precursors can induce 2 types of

murine T-ALL, i.e.. early immature T-ALL (*hhex*, *Mycn* and *lyl1* positive; these would be still TdTomato positive in this model) as well as mature murine T-ALLs (these would be TdTomato negative in this model). It might be interesting to also look at the TdTomato positive tumors in this model and see if the pattern of mutations, and the expression profile indeed is different.

We apologize for this inadvertent incorrect sentence. We have modified our language to better address this critique. In the revised version, this sentence has been replaced by “all *Sca1-Lmo2* T-ALL cases studied (18 out of 21)...”, because we could not do tomato FACS analysis in a few mice found death.

3. *Unexpectedly, the authors show that LMO2 activation in B-lineage also results in T-ALL (lower frequency). My main concern with these experiments would be potential leakage of these Cre Lines in early hematopoietic precursor cells or in the T-Lineage. Can the authors rule out that there might be any leakage associated with these lines. For example, with the AID-Cre line, it has been shown that this might be the case.*

Regarding the *Mb1-Cre* experimental approach, FACS analysis confirmed uniform and efficient GFP expression at the pro-B stage, and therefore all subsequent stages of B-cell differentiation (**Appendix Fig. S9A**). B cells from *Rosa26-Lmo2+Mb1-Cre* mice showed a developmental pattern comparable to that of B cells from their control littermates (**Appendix Fig. S9B**), which indicated that induction of *Lmo2* at the pro-B cell stage has a minimal effect on B-cell development. In the revised version we now show that GFP expression was not detected outside B-cell lineage as the frequency of GFP+ cells within both the BM myeloid progenitors and thymus T-cells was undetectable (**two new panels within Appendix Fig. S9A**). Similarly and regarding the *Aid-Cre* experimental approach, FACS analysis confirmed uniform and efficient GFP expression at GC stage (**Appendix Figure S9D**) and therefore within the subsequent stages of B-cell differentiation (new **Appendix Figure S9E**). In the revised version we also now show that GFP expression was not detected in bone marrow progenitor B cells, bone marrow myeloid cells and thymus T-cells from preleukemic *Rosa26-Lmo2+Aid-Cre* mice (new **Appendix Figure S9E**). Overall, these experiments dismiss a leakage of the Cre lines. In addition, the presence of BCR rearrangements proves the B-cell origin of T-cell tumors initiated in a B-cell.

4. *Throughout the manuscript, the FACS gates are not kept consistent in the comparison between WT and TG animals. For example, Suppl Fig 2. Please correct throughout the manuscript.*

We apologize for this error. FACS data have been reanalysed with the exact same gating throughout the manuscript in the revised version.

5. *Suppl Fig1C: A double peak is visible for GFP. Please explain*

We do not have an explanation for this double peak and it is correspond to a single population in terms of size and complexity or intensity of marker expression. An example is shown below with the B220+IgM+ population of periphery blood of a preleukemic *Rosa26-Lmo2+Sca1-Cre* mouse. This GFP expression is specific of the *Rosa26-Lmo2* mice and it is not present in wild-type mice. We only see this

double peak with the *Rosa26-Lmo2* mice, suggesting that the double peak might be dependent on Lmo2 expression.

6. Fig5C + Supp Fig 5C: How do the authors explain that in the preleukemic situation, there are mature CD4+ as well as mature CD8+ T cells that are positive for TdTomato? I guess the *sca1* promoter is not functional in these cells? Please explain.

Examination of the transgene expression pattern within various mouse hematopoietic lineages was performed by Miles et al (Development 124, 537-547, 1997), showing that the transgene is expressed in a significant fraction of CD4+ and CD8+ T cells within the thymus, respectively.

7. Figure 3A. What cells are used as control population for the comparison with the signature from the tumors? It is important to use the appropriate normal T cell control in these experiments.

In methods section of the original manuscript, we mentioned that "*Tumoral and normal thymuses were harvested from 10 Sca1-Lmo2 mice and 4 control littermate WT mice, respectively*". Now in the revised version we also indicate in Figure 3 legend (page 29) that "the cells that we used as control population for the comparison with the signature from the tumors were normal thymus T cells".

8. What type of tumours were generated in *nu/nu* mice? What was their immunophenotype?

The *Sca1-Lmo2+nu/nu* mice developed acute leukemias immunophenotypically distinct with co-expression of myeloid markers mimicking the human Early T-cell Precursor (ETP) phenotype. Consistently, the *Sca1-Lmo2+nu/nu* leukemias are transcriptionally related to human ETP leukemias. **These results are now included as new figure 5D.**

Referee #2:

T-ALL accounts for 15% of all childhood acute leukaemia cases. Here, TAL1 complexes with E2A/HEB binding subsequently to RUNX3, LMO1/2 and GATA3 forming a transcriptional scaffold complex that binds to the intergenic/intronic enhancer regions. The first two hits in T-ALL are supposedly TAL1 and LMO1/2 activation by different mutational events forming a chromatin regulatory circuit or looping that drives oncogene expression, which is well described in the literature. The third hit is usually hyperactivation of NOTCH signalling driving oncogenic MYC amplification that further invades the chromatin regulatory circuit boosting further reprogramming of malignant T-cells. The cell of origin of T-ALL was also described to be an immature thymocyte that originates from different stages of blocked early thymic development, where both the cortical as well as the subcapsular zone of the thymus for T-cell developmental stages were shown to be involved. How frequent a committed B-cell retro-differentiation program in human T-ALL would be relevant remains enigmatic. Still, this transgenic mouse model work is an elegant illumination of a key role of LMO2 as a genetic gardener to shape the transcriptome and even somatic mutation landscape of T-ALL. It is known since a long time that LMO2 is one of the most frequently mutated genes in T-ALL and the

authors describe it here with a transient expression by CRE-mediated recombination to be a main transcriptional reprogramming transcription factor that initiates a unique transcriptome of T-ALL. It even has the power and capacity similar like Pax5 loss to switch from a B-cell fate to a T-cell neoplasia, here the authors could discuss that part of the work better in light of Dr.'s Meinrad Busslinger and Stephen Nutt works which according to the reviewer opinion is a rare but another such interesting example for T-cell neoplasia etiology. Similar findings were e.g. also described in colorectal cancer by retro-differentiation by the Dr. Greten lab (Schwitalla et al., Cell, 2013) of epithelial differentiation being reprogrammed to intestinal stem cells by hyperactive cytokine and WNT signalling. Thus, the paper overall is well carried out, has a wealth of model work and genetic analysis, where only one major point and a few minor points could be performed to improve the impact of the study to the field of T-ALL.

We greatly appreciate this kind appraisal by Reviewer #2, and the thoughtful comments below which have significantly strengthened the revised manuscript. Now we have carefully addressed all comments in the revised manuscript as detailed below:

Major:

1) A true strength of the study is the somatic mutation and gene copy number gain or deletion analysis and that four different models for T-ALL were developed. However, a conclusion from the microarray analysis to similarity with human T-ALL is not justified. Better and more state-of-the-art is for sure an RNA-seq expression profile where e.g. the most significantly 500 genes from human T-ALL up- or down-regulated are compared to the murine T-ALL models used in this study. This heat map analysis would justify better for a conclusion. Thus, RNA-seq analysis should be performed on 3-4 controls, 3-4 T-ALL samples of genetic model 1 (Figure 1) and 3-4 T-ALL samples of genetic model 2 (Figure 2). If the authors can, one of the two B-cell identity driven third models for T-ALL developed in this study could also be profiled by RNA-seq to allow for a better conclusion if that also matches closely the transcriptome of human T-ALL or not and how well comparable it might be to their other more solid genetic models due to penetrance and latency. The nude mouse model work in Figure 4 is mechanistically interesting, but probably would not mimic closely a situation in patients with the exception of rare cases of SCID phenotype patients developing T-ALL in association with gene therapy as introduced by the authors and RNA-seq profiling here is not critical. Overall, a comparative RNA-seq analysis would be a true reality check with human T-ALL and a side-by-side comparison with conclusion of their best established model is a key finding. This could be a very strong Figure and conclusion for the work performed is then more transparent to the reader. Unfortunately as the data were recorded, current analysis cannot be judged which model is better or closest to human T-ALL, which would be advisable to be incorporated by simple RNA-seq analysis in this study. An alternative could be to compare their Affymetrix data to human Affymetrix data in similar reasoning as outlined above.

We want to thank the reviewer for this valuable suggestion and completely agree with his/her view that a comparative RNA-seq analysis would be a true reality check with human T-ALL. We have now performed RNA-seq analysis of depicted

mouse models and have added a novel Figure 6 A and 6B and a paragraph to the results section to the revised version of the manuscript.

To elucidate the differential transcriptomics landscape among different mouse models employed in this study, we performed paired-end RNA seq on *Rosa26-Lmo2* + *Sca1-Cre* (n=3), *Sca1-Lmo2* (n=4), *Rosa26-Lmo2+Mb1-Cre* (n=2), *Rosa26-Lmo2+Aid-Cre* (n=1) and WT-thymus (n=4) mice. The 500 genes with the highest variance among the difference murine models were depicted (**Fig. 6A**) with their corresponding FPKM values (Table EV5). Next Gene Set Enrichment Analysis (GSEA) of *Rosa26-Lmo2* x *Sca1-Cre* and *Sca1-Lmo2* mouse-based gene signatures, against a human T-ALL childhood expression set with healthy controls (Nat Genet 2003, 34, 267-273 and PNAS 2005, 102, 15545-15550) were performed (**Fig. 6B**). The up-regulated *Rosa26-Lmo2+Sca1-Cre* signature shows a significant enrichment in the human T-ALL group, which is in accordance to the human T-ALL situation wherein the expression of LMO2 is present throughout in tumor cells.

2) One important aspect and weakness of the paper is the lack of proof for presence or absence of LMO2. It would need to be a true controlled proof both for mRNA and protein data for LMO2 loss after T-ALL phenotypes have been established to allow for justification of the claim. The tomato-reporter data and immunohistochemistry in Figure 2 are not conclusive. Thus, the author should provide from sorted neoplastic cell populations of T-ALL where they make the claim that LMO2 is indeed lost with appropriate controls real-time mRNA expression quantification and direct Western blotting for LMO2 protein. This is particularly concerning where the authors make microarrays and write "...Cells were not sorted prior to RNA extraction for this analysis...." But cells are heterogenous/mosaic from the mice with and without LMO2 expression as clearly evidenced by reporter gene expression. Here, the authors need to give more detailed methods and re-write the results based on which cell populations were profiled. Certainly, the major purified neoplastic cell population is of most interest here and data could be viewed "contaminated" by unwanted cell types. The claim of LMO2 presence or absence needs then to be matched with sorted populations and a purity check on neoplastic cells that were sorted is mandatory to control for the procedure and to characterize the genetics of the models which is diverse and from text result body description classified as homogenous.

Following the reviewer advice, we have measured expression of *Lmo2* by both real-time PCR and western-blot in sorted-purified leukemic *Sca1-Lmo2* cells. Both approaches show that *Lmo2* expression is lost. **These results are now included as new Figure 2D-E.**

Despite *LMO2* being a T-cell-specific oncogene, *Lmo2* is not expressed in normal T-cells and functional *Lmo2* is not required for normal T-cell development (Mol Cell Biol. 2003 Dec; 23(24): 9003–9013). For this reason, we do not purify leukemic cells prior to microarray studies as the percentage of blast cells in the thymus was higher than 80%.

3) The gating performed in the FACS blots is often not controlled, the gates are touching the X- and/or Y-axis, which should never be the case, FACS data have to be reanalysed in improved and better controlled way, where e.g. a wildtype thymus

control has four gates for DN and DP to CD4 or CD8 single positive stages (one can e.g. look at Figure 1/2 on the gating of wt thymus compared to the experimental mice thymi) then the exact same gating shall be applied throughout the manuscript.

We apologize for this error. FACS data have been reanalysed with the exact same gating throughout the manuscript in the revised version.

4) Here, the authors use six tables to emphasize on their genetic findings, but a key strength would be to link similarities to human T-ALL simply by referencing. Thus, the authors could reduce the number of Tables to a more comprehensive format and add where possible human mutational or genetic gene loss or amplification literature findings. This would allow for a better description of similar findings in human T-ALL. Such a comprehensive Table can facilitate the understanding of the reader that model x is most similar to human T-ALL findings with genetic key findings as established here in their paper. Many cross-references are already in the text body and if reference limitations are an issue then all references for that comprehensive table could be also moved to a place in the paper with assistance of the editorial team.

Following the reviewer's advice, we have replaced the previous six tables by a single comprehensive Table that combines both the genetic characteristics of all mouse models presented in the manuscript with genetic figures of human LMO2 T-ALL presented in our paper close to genetic figures published in *Liu Y et al. Nat Genet. 2017 Aug;49(8):1211-1218.*

Minor:

1) The abstract contains unnecessary overstatements right from start and should be improved. Several wrong assumptions are given. T-ALL is from cancer genome landscape and cell fate decisions pretty well described and understood. See e.g. reviews by A. T. Look or S. Armstrong dating back to 2005 or newer ones.

In the revised version of the manuscript we have revised the abstract and avoid overstatements.

2) To description of Appendix Fig. S1A-B: it should be clarified that In the absence of Cre, neither Lmo2 nor eGFP are expressed, due to a STOP cassette in front of Lmo2 that has to be removed by CRE recombinase action to allow for their expression.

Following the reviewer advice, in the revised version of the manuscript we have clarified in Appendix Fig. S1A-B that "In the absence of Cre, neither Lmo2 nor eGFP are expressed, due to a STOP cassette in front of Lmo2 that has to be removed by Cre recombinase action to allow for their expression".

3) Certainly, it should be critically discussed that providing foreign transgenes to mice will raise unwanted immunogenicity in case of fluorescent protein expression or reporter usage that will be an issue.

We thank the reviewer for his/her comment. In pages 8-9 of the revised version we now mention that there is evidence to suggest that the immunogenicity and cytotoxicity of the fluorescent marker potentially may confound the interpretation of *in vivo* experimental data (Ansari AM, et al. Stem Cell Rev. 2016 Oct;12(5):553-559). We formally excluded the possibility that the cells that were originally marked with the fluorescent marker cannot be accurately traced over time by showing lack of Lmo2 expression by three different complementary approaches: immunohistochemistry (in the original manuscript) and by both real-time PCR and western-blot in sorted-purified leukemic *Sca1-Lmo2* cells in the revised version.

4) The CRE used in a subset of studies is a Sca1-Cre and Sca1 is a downstream target gene of STAT3 action, which could be cited. This would suggest that at least also more committed myeloid or lymphoid progenitor cell populations and not "stem cell" restricted cells recombine, since cytokine-driven STAT3 was described to drive SCA1 in mice also in more committed or differentiated cell types and not as suggested only in stem cells or HSC. This would be too vague or an overstatement, if at all then a restricted hematopoietic progenitor expression due to Sca1-CRE action might be concluded.

The Sca-1 hematopoietic stem cell marker is encoded by the allelic Ly-6E.1 and Ly-6A.2 genes (van de Rijn et al., 1989), which are members of the Ly-6 family consisting of at least 18 highly related genes (Kamiura et al., 1992). The Ly-6E.1 and Ly-6A.2 genes contain four exons that encode 876 and 830 base transcripts, respectively, and a 10-12 kDa GPI-linked cell surface glycoprotein (LeClair et al., 1986; McGrew and Rock, 1991; Palfree and Hammerling, 1986; Rock et al., 1986; Su and Bothwell 1989). The proteins are identical in sequence except for two amino acid differences resulting from three nucleotide differences (LeClair et al., 1986; Reiser et al., 1988). Ly-6A is the target for STAT3 but in this manuscript, we used the *Ly-6E* cassette to drive Cre expression. This allele is known to target HSC (Miles et al. Development 124, 537-547, 1997) and when we crossed Sca1-Cre with Rosa26-Lmo2 mice, all hematopoietic cells are GFP+ (Appendix Fig. S1C), indicating that a HSC is the cell-of-origin in this model.

5) The discussion could benefit from a few more points that could be relevant to bring the authors conclusions: Here, main focus was given on the potential of LMO2 to reprogram the T-cell lineage and it was described to be one of six key factors able to initiate induced HSC reprogramming, being also involved in B-cell neoplasias. The data suggest a more general role for LMO2 to shape the epigenome or to be involved in chromatin remodelling early on in T-ALL disease. It is not clear for the reviewer if in human T-ALL activation of LMO2 has to be switched off or if lower expression of LMO2 can be found in T-ALL. Also the second family member LMO1 might be similar in that regard to LMO2 or not, which should be discussed better.

We want to thank the reviewer for this valuable suggestion and completely agree with his/her view. The Discussion has been revised accordingly. To our knowledge, there is not available information about the level of LMO2 expression in human TALL.

6) Abbreviations should be defined at first use, examples are DLBCL or BCP-ALL, etc.

In the revised version, abbreviations have been defined at first use.

7) Fig. 1B Western is of poor quality and should be replaced.

The western of Figure 1D has been replaced in the revised manuscript.

8) Statistics are incomplete in Appendix Fig. S1E, since the diseased cells were not incorporated, if it is not significant that should be written, but it looks not to be analysed.

Appendix Fig. S1E shows the comparison of T cells populations of preleukemic Rosa26-Lmo2+Sca1-Cre mice versus control wild-type littermates. In the revised version we now indicate in Appendix Fig. S1E legend that "Preleukemic Rosa26-Lmo2+Sca1-Cre mice only show significant increase in CD4+ T-cells (p value=0.0286; Mann-Whitney test)"

9) The sentence in page 12 makes little to no sense where it says:.....Malignant T-cells were primarily CD8+/-CD4+/-.....at the end it is not the reader who can gamble what kind of cells the authors might refer too, in the Figure they claim the diseased cells are all double positive for CD4/CD8 and that could be a wrong description. Figure 5B has also single positive cells if one really looks at brightness of the cells.

We apologize for this inadvertent incorrect sentence. We have modified our language to better address this critique. In the revised version, that sentence has been replaced by "malignant T-cells were either double positive for CD4/CD8, or single positive for CD8 or single positive for CD4". Figure 5B has been revised accordingly.

2nd Editorial Decision

24th April 2018

Thank you for submitting your revised manuscript for consideration by The EMBO Journal. Your revised study was sent back to both referees for re-evaluation, and we have received comments from both of them, which I enclose below. As you will see the referees find that their concerns have been sufficiently addressed and they are now broadly in favour of publication.

Thus, we are pleased to inform you that your manuscript has been accepted in principle for publication in The EMBO Journal, pending some minor issues regarding material & methods and formatting as outlined below, which need to be adjusted at re-submission.

REFEREE REPORTS

Referee #1:

I would like to congratulate the authors with the revision of the manuscript. All my comments and concerns have been addressed. I have no further suggestions.

Referee #2:

The authors did a very careful job to address all concerns and to improve the manuscript significantly, so the study is significantly improved and it has a more simpler presentation to allow the reader to catch the new and very relevant conclusions. One could congratulate the authors for improvements during their revision work.

Corresponding Author Name: Dr. Isidro Sanchez-Garcia

Journal Submitted to: EMBOJ

Manuscript Number: EMBOJ-2017-98783R1